# Adaptive Data Collection for Robust Learning Across Multiple Distributions

**Chengbo Zang** [1]  **Mehmet Kerem Turkcan** [2]  **Gil Zussman** [1]  **Zoran Kostic** [1]  **Javad Ghaderi** [1]

## Abstract

We propose a framework for adaptive data collection aimed at robust learning in multi-distribution scenarios under a fixed data collection budget. In each round, the algorithm selects a distribution source to sample from for data collection and updates the model parameters accordingly. The objective is to find the model parameters that minimize the expected loss across all the data sources. Our approach integrates upper-confidence-bound (UCB) sampling with online gradient descent (OGD) to dynamically collect and annotate data from multiple sources. By bridging online optimization and multi-armed bandits, we provide theoretical guarantees for our UCB-OGD approach, demonstrating that it achieves a *minimax regret* of $O(T^{\frac{1}{2}}(K \ln T)^{\frac{1}{2}})$ over $K$ data sources after $T$ rounds. We further provide a lower bound showing that the result is optimal up to a $\ln T$ factor. Extensive evaluations on standard datasets and a real-world testbed for object detection in smart-city intersections validate the consistent performance improvements of our method compared to baselines such as random sampling and various active learning methods.

## 1. Introduction

In modern deep learning systems, sufficient and high-quality data is essential for robust model performance (Hestness et al., 2017). Although numerous standard datasets and pre-trained models are publicly available, they could fail to meet the diverse and specific requirements of applications, especially when applied to novel or previously unseen scenarios. Consequently, many applications–such as vision-language modeling (Laurençon et al., 2024), intelligent monitoring in healthcare (Moody & Mark, 1992; Zang et al., 2023), and object detection in smart cities (Cordts et al., 2016; Turkcan et al., 2024)–necessitate the collection and annotation of custom datasets to address the unique characteristics of their respective problem.

As a motivating example in smart-city applications, consider the task of vehicle detection at an urban traffic intersection. The objective is to develop a robust vehicle detection model that is capable of operating effectively under varying conditions, such as changes in lighting, occlusions, and weather variations. Three strategically placed cameras, each providing a unique perspective of traffic flow, are available for data collection. The trained model will be deployed across all three cameras, with the goal of optimizing the worst-case detection performance among them. However, annotating data for complex tasks such as detection, tracking, and segmentation is particularly expensive. This involves meticulous labeling of bounding boxes, object identities across frames, and pixel-level masks to generate accurate ground truths. Given a limited annotation budget (e.g. 2,500 images), it is crucial to strategically allocate the annotation budget across the three cameras to maximize the worst-case detection performance at the intersection.

In this paper, we present a framework for adaptive data collection and model training in multi-distribution scenarios under a fixed data annotation budget. The proposed framework operates iteratively, alternating between data collection (annotation) and model optimization in each round. Our objective is to devise a budget allocation strategy across the distribution sources such that the trained model achieves performance guarantee across *all* the distributions.

### 1.1. Related Work

**Active Learning.** The challenge of data annotation has driven significant advancements in the field of active learning (AL). The key idea of AL is to let the learning algorithm interactively query an annotator to label a subset of data points from a set of unlabeled data (Settles, 2009). In particular, *pool-based* methods assume that a pre-existing pool of unlabeled data is available and aim to select the most relevant samples from the pool to query for their labels. The relevance of a sample is often determined by criteria such as uncertainty measure (Lewis & Catlett, 1994) or

[1]Department of Electrical Engineering, Columbia University, New York NY, USA [2]Department of Civil Engineering, Columbia University, New York NY, USA. Correspondence to: Chengbo Zang <cz2678@columbia.edu>, Javad Ghaderi <jg3465@columbia.edu>.

*Proceedings of the 42nd International Conference on Machine Learning*, Vancouver, Canada. PMLR 267, 2025. Copyright 2025 by the author(s).

committee votes (Seung et al., 1992). On the other hand, *stream-based* methods observe a consecutive stream of samples and decide for every sample whether to query for its label or discard it. A similar branch of work is *active class selection*, where the learner is allowed to query a known class label for new samples (Lomasky et al., 2007; McClurg et al., 2023). AL methods are widely applied to Deep Neural Networks (DNNs) (Ren et al., 2022) in tasks such as classification (Ranganathan et al., 2017; Yoo & Kweon, 2019; Sinha et al., 2019) and object detection (Aghdam et al., 2019; Feng et al., 2019; Choi et al., 2021).

Despite their empirical effectiveness, AL methods are restricted by the quality of the generated queries. In a scenario with a trivial initial sample pool or a biased initial model, AL algorithms can exhibit unstable behavior by overfitting to a specific region of the data space or exacerbating the initial bias (Baldridge & Palmer, 2009; Karamcheti et al., 2021). Moreover, the existing theoretical analysis of AL is mainly restricted to linear hypothesis classes or basic problem setups like binary classification (Dasgupta, 2005; Wang et al., 2021; Gentile et al., 2022). Whereas some studies have derived coarse sample complexity bounds (Dasgupta, 2005) or analyzed convolutional neural networks using coreset techniques (Sener & Savarese, 2018), general theoretical guarantees remain elusive when it comes to more complex problem setups, such as those involving DNNs.

**Estimating the Dataset Size.** The relationship between DNN performance and the amount of available training data can be empirically characterized by the *neural scaling law* (Bisla et al., 2021; Hestness et al., 2017; Mahmood et al., 2022a). As a result, recent work models DNN training as a Markov Decision Process (Mahmood et al., 2022b) or a Gaussian Process (Tejero et al., 2023) with respect to (w.r.t.) the dataset size. The amount of data required given a specific performance metric can therefore be *empirically* predicted, although the composition of multiple data sources is often not explicitly accounted for.

The theoretical guarantees on dataset size can also be established by leveraging predefined data quality metrics, such as information functions (Xu & Zheng, 2017), submodular functions (Akcin et al., 2023a; Mirzasoleiman et al., 2016), or a target data distribution (Akcin et al., 2023b). Many centralized AL algorithms (e.g. uncertainty- or entropy-based sampling) share similar intuitions by selecting the most relevant data to annotate using such metrics. However, the robustness of real-world applications are often measured in terms of the worst-case model performances rather than the quality of the data itself. Therefore, it is more common in robust learning to directly minimize the worst-case loss, which we mainly discuss in this paper.

**Robust Learning and Multi-Armed Bandits.** Robust learning focuses on model generalization under distribution

shifts during training and testing (Ahuja et al., 2020). Specifically, *distributionally robust optimization (DRO)* formalizes this by minimizing the worst-case loss over a pre-defined uncertainty set of distributions, often characterized via metrics like Wasserstein distance or $f$-divergence (Duchi & Namkoong, 2021; Agarwal & Zhang, 2022). In such spirit, *group-DRO* explicitly incorporates group annotations to ensure uniform performance across subgroups, and utilizes *bandit algorithms* to address robustness and fairness (Haghtalab et al., 2022; Zhang et al., 2023).

Particularly, Multi-Armed Bandit (MAB) studies a sequential decision problem that seeks to maximize cumulative reward over time, where the action at each time step is selected from multiple fixed choices with unknown reward distributions (Robbins, 1952; Gittins, 1979). Popular algorithms, such as $\epsilon$-Greedy and UCB, have been extensively studied and analyzed for *stochastic bandits* (Auer et al., 2002). While *adversarial bandits* (Bubeck & Nicolò, 2012) are typically utilized in group-DRO literature, such algorithms often disregard the notion of a dataset by considering an oracle-based setup, where the algorithm samples directly from the data distribution to obtain an unbiased loss (and gradient) estimator. Despite its theoretical convenience, directly sampling from the data distribution every time is not always feasible due to practical limitations. In contrast, this work seeks to utilize the information contained in the collected dataset by leveraging algorithms from stochastic bandits and perspectives from *contextual bandits* (Langford & Zhang, 2007; Slivkins, 2011).

### 1.2. Contributions

Our main contributions can be summarized as follows.

- We introduce an adaptive data collection framework for robust learning across multiple distributions under a limited data collection and annotation budget, without relying on an initially collected set of annotated or unannotated samples.

- We propose the UCB-OGD algorithm that combines UCB sampling and online gradient descent which achieves $O(T^{\frac{1}{2}}(K \ln T)^{\frac{1}{2}})$ minimax regret, matching the theoretical lower bound up to a $\ln T$ factor.

- We conduct experiments on both standard datasets and a real-world testbed for complex tasks and demonstrate that the proposed UCB-OGD algorithm achieves comparable or higher minimax performance on multiple tasks compared to well-known AL algorithms.

## 2. Problem Statement

**Notations.** We consider a data space $\mathbb{X}$ with $K$ data sources and a parametrized model we wish to train for some task. This can be classification, multi-class object detection and

segmentation, or even single-class tasks where the data can be obtained from different sources (e.g. see our motivating smart-city example in Section 1). Each data source $k \in \{1, 2, \ldots, K\}$ is associated with an unknown data distribution $\mathcal{D}_k$ over $\mathbb{X}$. Let $\theta$ be the parameter of the trainable model in some parameter space $\Theta$. Let $\ell(\theta, X)$ be the loss function for data point $X \in \mathbb{X}$[1]. Let $\mu_k(\theta) := \mathbb{E}_{X \sim \mathcal{D}_k}[\ell(\theta, X)]$ denote the expected loss associated with data source $k$, and $\nabla \mu_k(\theta) := \mathbb{E}_{X \sim \mathcal{D}_k}[\nabla_\theta \ell(\theta, X)]$ be the gradient of $\mu_k(\theta)$ w.r.t. the parameter $\theta$.

For instance, in the motivating smart-city example in Section 1, there are $K = 3$ data sources, one for each camera, $X$ is an image randomly obtained from a camera with corresponding data source index, object classes, and bounding boxes, $\theta$ is the parameter vector of an object detection model, and $\ell(\theta, X)$ is the loss of the model prediction over the input $X$ given its annotation.

Let $\mathcal{S}$ be a set of samples and denote $\mathcal{S}_k := \{X \in S : X \sim \mathcal{D}_k\}$ as the subset of $\mathcal{S}$ that belongs to data source $k$. Then the *empirical estimate* of $\mu_k(\theta)$ over the set of samples $\mathcal{S}$ can be computed as $\hat{\mu}_k(\theta; \mathcal{S}) := \sum_{X \in \mathcal{S}_k} \ell(\theta, X) / |\mathcal{S}_k|$, where $|\cdot|$ denotes the set cardinality. Similarly, the empirical estimate of $\nabla \mu_k(\theta)$ over the set of samples $\mathcal{S}$ is computed as $\nabla \hat{\mu}_k(\theta; \mathcal{S}) := \sum_{X \in \mathcal{S}_k} \nabla_\theta \ell(\theta, X) / |\mathcal{S}_k|$.

**Optimization Objective.** If $\mathcal{S}$ is a fixed training set that exists in advance, a natural objective is to minimize the empirical loss over the training set, i.e., $\sum_{X \in \mathcal{S}} \ell(\theta, X) / |\mathcal{S}|$ (a.k.a. *empirical risk minimization*). This objective function can be interpreted as the weighted average of the empirical losses of all data sources, where each data source is weighted by the ratio of its samples in $\mathcal{S}$, i.e., $\sum_{k=1}^{K} (|S_k| / |S|) \hat{\mu}_k(\theta; \mathcal{S})$. However, the construction of the training set itself may worth more careful considerations, especially in real-world applications where the annotation budget is limited. As opposed to allocating the budget of training samples among the data sources in a predefined way (e.g. uniformly), we allow the samples to be actively collected and annotated during the training process. In this scenario, the requirement of a preexisting training set $\mathcal{S}$ is alleviated. Since no prior exists for the ratio $|\mathcal{S}_k| / |\mathcal{S}|$, we consider an objective that is independent of this ratio through *minimax* optimization:

$$\min_\theta \ \max_{k=1,\ldots,K} \ \mu_k(\theta). \tag{1}$$

Note that the expected loss functions $\mu_k(\theta)$, for $k = 1, \ldots, K$, are *unknown*. The objective (1) is of particular interest for the purpose of optimizing data collection. It focuses on optimizing the *worst-case* expected loss which ensures the fairness of the algorithm (Papadaki et al., 2022)

[1]To be precise, $X = (x, y)$ where $x$ is the input, and $y$ is the target output (annotation) from the model with input $x$. Then $\ell(\theta, X)$ measures how different the prediction $\hat{y}$ of the model with parameter $\theta$ is from the target output $y$.

---

**Algorithm 1** General Framework of Online Optimization with Adaptive Data Collection

**Require:** Total training rounds $T$, batch size $M$, randomly initialized $\theta_1$
1: $\mathcal{X}_0 \leftarrow \varnothing$
2: **for** $t = 1, 2, \ldots, T$ **do**
3:     $k_t \leftarrow \text{SELECT}(\theta_t, \mathcal{X}_{t-1})$
4:     $\mathcal{B}_t \leftarrow \{X_1, \ldots, X_M \sim \mathcal{D}_{k_t}\}$
5:     $\mathcal{X}_t \leftarrow \mathcal{X}_{t-1} \bigcup \mathcal{B}_t$
6:     $\theta_{t+1} \leftarrow \text{UPDATE}(\theta_t, \mathcal{X}_t, k_t)$
7: **end for**

---

and is less prone to overfitting towards a particular data source. Minimax learning is also preferred for its robustness to distributional uncertainties (Farnia & Tse, 2016), since we want to train a model that works well across a range of data distributions one might encounter during real-world deployment.

**Algorithm Framework.** We propose a general framework by combining adaptive data collection and online optimization as presented in Algorithm 1. The algorithm starts with an empty training set $\mathcal{X}_0 = \varnothing$ and an initialized $\theta_1$. In every round $t$, in the SELECT step, the algorithm selects a data source $k_t \in \{1, 2, \ldots, K\}$ to *collect and annotate* a batch of samples $\mathcal{B}_t$. The decision is made based on the current model parameter $\theta_t$ and the existing training set $\mathcal{X}_{t-1}$. The batch of new samples $\mathcal{B}_t$ is then added to $\mathcal{X}_{t-1}$ to get the updated training set $\mathcal{X}_t$. For the simplicity of the analysis, we assume the batch size is fixed, i.e., $|\mathcal{B}_t| = M \geq 1$. Then, in UPDATE step, the algorithm updates the model parameter $\theta_t$ based on the updated training set $\mathcal{X}_t$ and the selected data source $k_t$, and obtains $\theta_{t+1}$ for the next round.

**Performance Metric.** After $T$ rounds of execution, an online algorithm $\mathcal{A}$ generates a sequence of data source indices $k_1, \ldots, k_T$ and a sequence of model parameters $\theta_1, \ldots, \theta_T$. A natural metric to quantify the performance of an online algorithm that solves the optimization problem (1) is based on the *minimax regret*, which is defined as the cumulated gap between the global optimal loss and the maximum loss achieved by the current model in each round.

**Definition 2.1** (Minimax Regret). The minimax regret of an algorithm $\mathcal{A}$ over $T$ rounds is defined as

$$R(\mathcal{A}_T) := \sum_{t=1}^{T} \max_k \mu_k(\theta_t) - T \min_\theta \ \max_k \mu_k(\theta), \tag{2}$$

where we use $\mathcal{A}_T = \{(k_t, \theta_t), t = 1, \ldots, T\}$ to denote the sequence of data source indices and model parameters generated by algorithm $\mathcal{A}$ after $T$ rounds.

We mainly focus on the expectation of the minimax regret, $\mathbb{E}[R(\mathcal{A}_T)]$, since the trajectory $\mathcal{A}_T$ is random.

The minimax regret metric can be connected to the convergence of *time-averaged model parameter* under algorithm $\mathcal{A}$, defined as $\bar{\theta}_{\mathcal{A}_T} := \sum_{t=1}^T \theta_t / T$. The time-average convergence is commonly adopted in the literature of online and stochastic algorithms, as it facilitates more straightforward theoretical guarantees (Hazan, 2016; Tejero et al., 2023). While there is no *direct* equivalence between the average parameter $\bar{\theta}_{\mathcal{A}_T}$ and the final parameter $\theta_T$ (which is a more common choice in actual implementations), advanced optimizers such as SGD with Momentum and Adam (Polyak, 1964; Kingma & Ba, 2017) run a moving average over the gradients $\nabla \hat{\mu}_k$ to improve the robustness of the algorithm. These two concepts are analogous in terms of promoting smoother update steps for the model.

When the loss functions are convex, we can build an intuitive relationship between the minimax regret and the *optimality gap* as follows.

**Proposition 2.2** (Optimality Gap). *Let $\mu_1, \ldots, \mu_K$ be convex in $\theta$. Then any algorithm $\mathcal{A}$ satisfies*

$$\mathbb{E}\left[\max_k \mu_k(\bar{\theta}_{\mathcal{A}_T})\right] - \min_\theta \max_k \mu_k(\theta) \leq \frac{\mathbb{E}[R(\mathcal{A}_T)]}{T}, \quad (3)$$

*where $\bar{\theta}_{\mathcal{A}_T} = \sum_{t=1}^T \theta_t / T$.*

The proof of Proposition 2.2 follows from the convexity of $\max_k \mu_k$ and application of Jensen's inequality, i.e., $\max_k \mu_k(\bar{\theta}_{\mathcal{A}_T}) \leq \sum_{t=1}^T \max_k \mu_k(\theta_t)/T$.

As a result of Proposition 2.2, to show that an algorithm converges to the minimax optimum, it is sufficient to show that its (expected) minimax regret is *sublinear* in $T$.

We adopt the following assumptions for the analysis presented in this paper.

**Assumption 2.3** (Bounded Lipschitz Loss). There exists some $C \geq 0$ s.t. $\ell(\theta, X) \in [0, C]$ for all $X$ and $\theta$. Also, the expected loss $\mu_k(\theta)$ is $L$-Lipschitz in $\theta$ for all $k$.

**Assumption 2.4** (Finite Domain). The model parameters generated by Algorithm 1 in all rounds, $\theta_1, \ldots, \theta_T$, lie in a bounded subset of $\Theta$ with diameter $D \geq 0$. [2]

**Assumption 2.5** (Finite Gradient Noise). There exists some $\sigma \geq 0$ s.t. the variance of the gradient is finite, i.e., $\mathbb{E}_{X \sim \mathcal{D}_k}[\|\nabla \ell(\theta, X) - \nabla \mu_k(\theta)\|^2] \leq \sigma^2$ for all $k, \theta$.

**Assumption 2.6** (IID Sampling). Data collected from every data source $k$ is sampled *i.i.d.* from the associated data distribution $\mathcal{D}_k$.

## 3. Algorithms and Main Results

We present three specific algorithms within the framework of Algorithm 1 and their corresponding performances. Since

we fix the batch size $M$ and the total number of rounds $T$, the algorithm collects a total number of $MT$ samples from all data sources, allowing for uniform comparisons between different algorithms based on their minimax regret.

For the optimization step (Line 6 of Algorithm 1), we consider *Online Gradient Descent* (OGD) (Hazan, 2016). Recall that $k_t$ is the data source selected for the current round $t$. Denote $\mathcal{X}_{t,k_t} := \{X \in \mathcal{X}_t : X \sim \mathcal{D}_{k_t}\}$ as the subset of $\mathcal{X}_t$ collected from data source $k_t$. Let $\mathcal{S}$ be a batch of data points uniformly sampled from $\mathcal{X}_{t,k_t}$. Then OGD updates the model parameter of the next round by taking a step in the direction of the estimated gradient of the mean loss of source $k_t$ at the current round, i.e.,

$$\theta_{t+1} \leftarrow \theta_t - \eta_t \nabla \hat{\mu}_{k_t}(\theta_t; \mathcal{S}), \quad (4)$$

where $\eta_t := 1/(2L\sqrt{t})$ is the learning rate.

For the data source selection step (Line 3 of Algorithm 1), we consider the following three methods.

**Random Selection.** The simplest baseline is to pick the data source uniformly at random, i.e., $k_t \sim \mathrm{U}(\{1, \ldots, K\})$[3]. This is equivalent to uniformly allocating the budget of $MT$ samples among the $K$ data sources, yielding approximately $MT/K$ samples per data source. We refer to Algorithm 1 with random selection and OGD as *Rand-OGD*.

Intuitively, Rand-OGD is not designed for the minimax objective in Equation (1), since all data sources are queried in a balanced way regardless of their losses (see Appendix A.4). A more viable selection method that addresses the minimax problem is to greedily select the data source that incurs the highest loss, i.e., $k_t \leftarrow \max_k \mu_k(\theta_t)$. However, the true expectation $\mu_k(\theta_t)$ is *unknown* and we can only measure its empirical estimate $\hat{\mu}_k(\theta_t; \mathcal{X}_{t-1})$ which is a random variable. Moreover, the deviation of $\hat{\mu}_k$ from its expectation can be particularly large with a small number of samples.

While we need to focus on optimizing the maximum loss associated with the data source that incurs it as much as possible (i.e., *exploitation*), we also need to ensure that enough samples are collected from other data sources in order to reduce the variance of the estimated losses (i.e., *exploration*). This resembles the exploration-exploitation trade-off in MAB (Multi-Armed Bandit) problems. We consider the following two data source selection methods inspired by MAB algorithms (Auer et al., 2002).

**Decaying $\epsilon$-Greedy Selection.** For $t > 1$, define an exploration probability $\epsilon_t$ as

$$\epsilon_t := \frac{1}{2} \sqrt[3]{\alpha K \ln t / (2M(t-1))}, \quad (5)$$

where $\alpha \geq 1/2$ is a constant. We specially define $\epsilon_1 := 1$. Then, in every round $t$, with probability $\epsilon_t$, we select $k_t \sim$

---

[2]This will be defined more rigorously in Appendix A.3.

[3]$\mathrm{U}(S)$ denotes the uniform distribution over set $S$.

$U(\{1, \ldots, K\})$, otherwise, we select

$$k_t \leftarrow \arg\max_k \hat{\mu}_k(\theta_t; \mathcal{X}_{t-1}). \quad (6)$$

We refer to Algorithm 1 with $\epsilon_t$-Greedy selection and OGD as *Eps-OGD*.

**Upper-Confidence-Bound (UCB) Selection.** UCB is another popular exploration-exploitation strategy in MAB that balances the the empirical estimate and its uncertainty. Define a confidence radius for each data source $k$ given some set of samples $\mathcal{S}$ as

$$r_k(\mathcal{S}) := C\sqrt{\alpha \ln t / (2|\mathcal{S}_k|)}, \quad (7)$$

where $C$ is defined in Assumption 2.3, $\alpha \geq 1/2$ is a constant, and $|\mathcal{S}_k|$ is the number of samples in $\mathcal{S}$ that belongs to data source $k$. Then, in every round $t$, we pick the data source with the maximum UCB value, i.e.,

$$k_t \leftarrow \arg\max_k \hat{\mu}_k(\theta_t | \mathcal{X}_{t-1}) + r_k(\mathcal{X}_{t-1}). \quad (8)$$

We refer to Algorithm 1 with UCB selection and OGD as *UCB-OGD*.

The following theorem states our main result regarding the minimax regret of Eps-OGD and UCB-OGD for convex loss functions.

**Theorem 3.1** (Minimax Regret). *Let $\mu_1, \ldots, \mu_K$ be convex in $\theta$. Then Eps-OGD and UCB-OGD achieve the following minimax regrets:*

$$\begin{aligned} \mathbb{E}[R(\text{Eps-OGD}_T)] &= O(T^{\frac{2}{3}}(K\ln T)^{\frac{1}{3}}) \\ \mathbb{E}[R(\text{UCB-OGD}_T)] &= O(T^{\frac{1}{2}}(K\ln T)^{\frac{1}{2}}) \end{aligned}. \quad (9)$$

Note that, by Proposition 2.2, we can subsequently conclude that the expected minimax optimality gap of Eps-OGD and UCB-OGD diminishes at the rate $O(T^{-\frac{1}{2}}(K\ln T)^{\frac{1}{2}})$ and $O(T^{-\frac{1}{3}}(K\ln T)^{\frac{1}{3}})$, respectively.

*Remark* 3.2. When the loss functions are *non-convex*, it is generally not feasible to converge to the global optimum of (1). In this case, we can only show convergence to a pareto-stationary point (Sener & Savarese, 2018). Formally, $\theta_s$ is called *Pareto Stationary* if there exists a set of $\alpha_1, \ldots, \alpha_K$ s.t. $\sum_{k=1}^{K} \alpha_k \nabla \mu_k(\theta_s) = 0$, where $\alpha_k \geq 0$ for all $k$ and $\sum_{k=1}^{K} \alpha_k = 1$. We can use time-smoothing w.r.t. a non-trivial window $1 \ll w \leq T$ and corresponding time-smoothed OGD algorithms from the online non-convex optimization (Hazan et al., 2017; Hallak et al., 2021). Then we can show that asymptotically, as $T, w \to \infty$, any time-smoothed OGD-based algorithm $\mathcal{A}$ converges to a pareto-stationary point $\theta_s$ where $\sum_{k=1}^{K} \alpha_k \nabla \mu_k(\theta_s) = 0$, and $\alpha_k$ is the fraction of rounds that data source $k$ is selected in the long run under $\mathcal{A}$. We provide the formal statement of this result and its proof in Appendix A.5 for completeness.

A natural question is whether the bounds in Theorem 3.1 can be improved. We can establish the following lower-bound for the minimax regret of any algorithm which shows UCB-OGD is optimal, up to a $\ln T$ factor.

**Proposition 3.3** (Minimax Lower-Bound). *The minimax regret of any online algorithm $\mathcal{A}$ satisfies $\mathbb{E}[R(\mathcal{A}_T)] \geq O(T^{\frac{1}{2}})$ in the worst case.*

The proof of Proposition 3.3 is based on a simple case and is provided in Appendix A.6.

## 4. Proof of Main Results (Theorem 3.1)

In Algorithm 1, both the SELECT step and the UPDATE step seek to utilize the information within the collected training set $\mathcal{X}_t$, rather than generating fresh samples from the data distribution (Haghtalab et al., 2022; Zhang et al., 2023). The intuition is that discarding previous samples will result in the model being trained on every data point *only once*, which is infeasible for most modern DL tasks such as object detection. Instead, it is conventional to reuse past samples while maintaining a training set, at the cost of potential complexities in generalization during theoretical analysis.

Formally, consider the empirical loss $\hat{\mu}_k(\theta_t; \mathcal{X}_t)$ for some $k, \theta_t$ and a training set $\mathcal{X}_t$. When $\theta_t$ is trained over the collected samples in $\mathcal{X}_t$, optimization steps like OGD in Equation (4) introduces implicit dependency between $\theta_t$ and $\mathcal{X}_t$, which makes the empirical loss estimator (thus the empirical gradient estimator) *biased*, i.e.,

$$\tilde{\mu}_k(\theta_t) := \mathbb{E}[\hat{\mu}_k(\theta_t; \mathcal{X}_t)|\mathcal{A}_t] \lesssim \mu_k(\theta_t). \quad (10)$$

Indeed, $\tilde{\mu}_k(\theta_t)$ tends to *underestimate* $\mu_k(\theta_t)$ due to potential overfitting. In practical DL training, this is mitigated empirically by techniques such as *data augmentation* and *regularization*, which we also adopt in the experiments in Section 5.

To characterize the minimax regret (Definition 2.1) of the algorithms, we present several standard regret definitions from the literature.

**Optimization Regret.** Recall that Algorithm 1 picks one data source $k_t$ in every round $t$ and generates a $\theta_{t+1}$ for the next round $(t + 1)$ based on the updated data set $\mathcal{X}_t$. After $T$ rounds of execution, we define an optimization regret for algorithm $\mathcal{A}$ based on the cumulated gap between the expected empirical loss $\tilde{\mu}_{k_t}(\theta_t)$ achieved by the algorithm in round $t$ and the best fixed model parameter $\theta$ chosen *in hindsight* given the sequence of data sources $k_1, k_2, \ldots, k_T$ (Hazan, 2016), i.e.,

$$R_o(\mathcal{A}_T) := \sum_{t=1}^{T} \tilde{\mu}_{k_t}(\theta_t) - \min_\theta \sum_{t=1}^{T} \tilde{\mu}_{k_t}(\theta). \quad (11)$$

**Bandit Regret.** Consider a $K$-armed *contextual bandit*, where the reward of each arm $k$ is $\ell(\theta, X)$, $X \sim \mathrm{U}(\mathcal{X}_{t,k})$ with expectation $\tilde{\mu}_k(\theta)$ under context $\theta$. In our problem, the arms are the data sources and the context $\theta$ is the model parameter. Further, the reward distribution of every arm-context pair is stationary[4]. After $T$ rounds, we define the bandit regret of an algorithm $\mathcal{A}$ based on the cumulated gap between the expected empirical loss when the arms are chosen optimally in a *context-aware* manner and the actual expected empirical loss achieved by the algorithm in each round (Slivkins, 2011), i.e.,

$$R_b(\mathcal{A}_T) := \sum_{t=1}^{T} \left( \max_k \tilde{\mu}_k(\theta_t) - \tilde{\mu}_{k_t}(\theta_t) \right). \quad (12)$$

**Generalization Regret.** We further define the generalization regret as the cumulated gap between the expected worst-case empirical loss and the worst-case true loss, i.e.,

$$R_g(\mathcal{A}_T) := \sum_{t=1}^{T} \left( \max_k \mu_k(\theta_t) - \max_k \tilde{\mu}_k(\theta_t) \right). \quad (13)$$

Using the definitions above, the minimax regret (Definition 2.1) can be decomposed as follows.

**Proposition 4.1** (Regret Decomposition). *Let Equation* (10) *hold. Then the minimax regret of any algorithm $\mathcal{A}$ over $T$ rounds satisfies $R(\mathcal{A}_T) \leq R_o(\mathcal{A}_T) + R_b(\mathcal{A}_T) + R_g(\mathcal{A}_T)$.*

*Proof.* By definition, we can write

$$R_o(\mathcal{A}_T) + R_b(\mathcal{A}_T) + R_g(\mathcal{A}_T)$$
$$= \sum_{t=1}^{T} \max_k \mu_k(\theta_t) - \min_\theta \sum_{t=1}^{T} \tilde{\mu}_{k_t}(\theta)$$
$$\geq \sum_{t=1}^{T} \max_k \mu_k(\theta_t) - \min_\theta \sum_{t=1}^{T} \mu_{k_t}(\theta) \quad . \quad (14)$$
$$\geq \sum_{t=1}^{T} \max_k \mu_k(\theta_t) - T \min_\theta \max_k \mu_{k_t}(\theta)$$
$$= R(\mathcal{A}_T)$$

The first inequality in Equation (14) follows from Equation (10), and the second inequality from the fact that $\mu_{k_t}(\theta) \leq \max_k \mu_k(\theta)$ for any $k_t, \theta$. $\square$

For the optimization regret $R_o$, we can establish the following result for both Eps-OGD and UCB-OGD.

---

[4]This means, at any two time steps $t_1, t_2 \in \{1, \ldots, T\}$, if $k_{t_1} = k_{t_2} = k$ and $\theta_{t_1} = \theta_{t_2}$, then $\ell(\theta_{t_1}, X_1)$, and $\ell(\theta_{t_2}, X_2)$ for $X_1, X_2 \sim \mathrm{U}(\mathcal{X}_{t,k})$ are *i.i.d.*

**Proposition 4.2** (Optimization Regret). *Let $\mu_1, \ldots, \mu_K$ be convex in $\theta$. With step sizes $\eta_t = 1/(2L\sqrt{t})$, the optimization regret of any OGD-based algorithm $\mathcal{A}$ over $T$ rounds satisfies*

$$\mathbb{E}[R_o(\mathcal{A}_T)] \leq L(D^2 + 1)\sqrt{T} + L^{-1}\sigma^2\sqrt{T}, \quad (15)$$

*where $L$, $D$, and $\sigma$ are defined in Assumption 2.3, Assumption 2.4, and Assumption 2.5, respectively.*

Proof of Proposition 4.2 follows from the analysis of standard OGD (Hazan, 2016) with modifications to account for Lipschitz loss function (Assumption 2.3) and the gradient noise (Assumption 2.5) in our stochastic setting. We also alleviate the dependence of the learning rate $\eta_t$ on the diameter $D$ (Assumption 2.4) to be more in line with the conventions in stochastic optimization literature (Garrigos & Gower, 2024). The complete proof of Proposition 4.2 is provided in Appendix A.1.

For the bandit regret $R_b$, our problem adopts the structure of a contextual bandit in a rigorous manner. However, we are *not* necessarily playing the bandit game here. When a new context $\theta_t$ arrives in round $t$, the player in a rigorous bandit game can only learn about the context from the specific problem structure (e.g. similarity information in the context space), or by pulling the arms for new samples. In contrast, both Eps-OGD and UCB-OGD in our case are allowed to learn about the new context $\theta_t$ *directly* by evaluating the loss over the collected samples, i.e., computing the empirical loss $\hat{\mu}_k(\theta_t|\cdot)$. This provides us with information about the new context $\theta_t$ even if no new samples are collected in the current round. We can take this advantage to bypass the challenge of navigating through arm-context pairs based on similarity and directly utilize the empirical loss as a more informative source of knowledge about $\theta_t$.

We sketch the outline of the bandit regret analysis below and provide the detailed proofs in Appendices A.2 and A.3. The core steps involve establishing the concentration inequality and characterizing the bandit gap.

**Lemma 4.3** (Loss Concentration). *Let $\mathcal{S} \sim \mathrm{U}(\mathcal{X}_t)$ be a batch of training data sampled uniformly at random from the training set $\mathcal{X}_t$. Then it holds for any constants $k \in \{1, \ldots, K\}$ and $r > 0$ that*

$$\Pr[|\hat{\mu}_k(\theta_t; \mathcal{S}) - \tilde{\mu}_k(\theta_t)| > r]$$
$$\leq 2 \exp \left( -\frac{2|\mathcal{S}_k|}{C^2} \cdot r^2 \right), \quad (16)$$

*where $|\mathcal{S}_k|$ denotes the number of samples in $\mathcal{S}$ that belongs to data source $k$, and $C$ is defined in Assumption 2.3.*

The proof of Lemma 4.3 follows from constructing a *martingale* over the sequence of $\ell(\theta_t, X_i)$ for each $X_i \in \mathcal{S}_k$ and applying the Azuma's inequality, detailed in Appendix A.2.

Using the tail bound provided in Lemma 4.3, now we show the concentration of the empirical loss *on expectation*, expressed in terms of a confidence radius. While the UCB algorithm directly defines its confidence radius in Equation (7), we similarly define the confidence radius of the $\epsilon$-Greedy algorithm given some set of samples $\mathcal{S}$ as

$$r_k(\mathcal{S}) := C\sqrt{\alpha K \ln t / (2|\mathcal{S}|\epsilon_t)}. \tag{17}$$

Further denote $\tilde{\mu}^\star(\theta_t) := \max_k \tilde{\mu}_k(\theta_t)$ the maximum expected empirical loss, and $r^\star(\mathcal{S})$ the corresponding confidence radius defined in Equations (7) and (17). Let $\hat{\mu}^\star(\theta_t; \mathcal{S})$ be the empirical estimate of $\tilde{\mu}^\star(\theta_t)$ given $\mathcal{S}$. The following is a direct consequence of Lemma 4.3.

**Lemma 4.4** (Bandit Gap). *Let* $\Delta_k(\theta_t) := \tilde{\mu}^\star(\theta_t) - \tilde{\mu}_k(\theta_t)$ *and* $\hat{\Delta}_k(\theta_t; \mathcal{S}) := \hat{\mu}^\star(\theta_t; \mathcal{S}) - \hat{\mu}_k(\theta_t; \mathcal{S})$. *Then for any constant* $\alpha \geq 1/2$ *and confidence radius* $r_k$ *defined in Equations* (7) *and* (17)*, it holds that*

$$\mathbb{E}[\Delta_k(\theta_t)|\mathcal{A}_t, \mathcal{S}] \leq r_k(\mathcal{S}) + r^\star(\mathcal{S}) \\ + \hat{\Delta}_k(\theta_t; \mathcal{S}) + O(t^{-\alpha}). \tag{18}$$

This indicates that, by following the decision rules of the bandit algorithms that carefully controls $\hat{\Delta}_{k_t}(\theta_t; \cdot)$ with appropriately chosen confidence radius, the estimation of the worst-case loss is accurate with *high probability*. Since the bandit regret in Equation (12) is an accumulation of the bandit gaps $\Delta_{k_t}(\theta_t)$ in each round, Lemma 4.4 effectively provides a provable probability bound on the concentration of the empirical estimation of the bandit gaps, which also guarantees the quality of the OGD updates.

The following results on the bandit regret match the standard regret bounds of stationary stochastic bandits (Auer et al., 2002) despite the non-stationary setting of our problem. The detailed proof is given in Appendix A.3.

**Proposition 4.5** (Bandit Regret). *The bandit regret (Equation* (12)*) is* $O(T^{\frac{2}{3}}(K \ln T)^{\frac{1}{3}})$ *for* Eps-OGD*, i.e.,*

$$\mathbb{E}[R_b(\text{Eps-OGD}_T)] \leq \frac{3(2\sqrt{2}+1)C}{2}\sqrt[3]{\frac{\alpha K T^2 \ln T}{2M}}, \tag{19}$$

*and* $O(N^{\frac{1}{2}}(K \ln T)^{\frac{1}{2}})$ *for* UCB-OGD*, i.e.,*

$$\mathbb{E}[R_b(\text{UCB-OGD}_T)] \leq 2C\sqrt{\frac{2\alpha K T \ln T}{M}}. \tag{20}$$

*Remark* 4.6. In the rigorous contextual bandit setting where one can only learn about the new context from the problem structure, it is common to assume that the reward function is Lipschitz w.r.t. to the context. The uniform partition algorithm (Hazan & Megiddo, 2007) proposes to partition the context space and run a stationary bandit algorithm on

every partition and incurs the regret $O(T^{1-\frac{1}{2+K+H}})$, where $H$ is the *covering dimension* of the context space $\Theta$. Since usually $H \gg 1$ for modern DNNs (Mao et al., 2024), the uniform partition algorithm and other similar contextual bandit algorithms (Slivkins, 2011) may struggle to obtain a meaningful regret in our setting.

The final step is to bound the generalization regret $R_g$, which has been studied extensively in robust learning literature (Dziugaite & Roy, 2017; Arora et al., 2018; Cao & Gu, 2019). While this is out-of-scope for the purpose of this paper, the takeaway is that the generalization bound takes the form of

$$\frac{R_g(\mathcal{A}_T)}{T} \leq \sqrt{\frac{\mathcal{C}(\theta)}{|\mathcal{X}_T|}} = O(T^{-\frac{1}{2}}), \tag{21}$$

where $\mathcal{C}(\theta)$ is a constant determined by the *complexity* of the model. Intuitively, the generalization is better when the size of the dataset is sufficiently large, and when the model is not *too* over-parametrized. This is indeed true for a wide range of complex DL tasks, including our motivating example of urban vehicle detection described in Section 1.

**Proof of Theorem 3.1.** The proof is a direct consequence of Proposition 4.1 and the fact that the optimization regret in Proposition 4.2, the bandit regrets in Proposition 4.5, and the generalization regret in Equation (21) are all sublinear.

## 5. Experimental Results

In this section, we present the experimental results of the three algorithms described in Section 3, compared to other state-of-the-art AL (active learning) algorithms. We consider the following tasks with different notions of *data source*, demonstrating the flexibility of our framework in practical settings.

**Classification.** We perform image classification on the CIFAR10 dataset (Krizhevsky et al., 2009) with a budget of 10,000 images, where every class is a data source. We also report the results on the MNIST dataset (Lecun et al., 1998) to test different optimizer configurations and get more insight into the distribution of collected samples from different classes under different algorithms. The metric are the mean and minimum class-wise accuracies among all classes. We use a simple three-layer convolutional neural network (CNN) architecture with ReLU activations for this task.

**Multi-class Object Detection.** We perform object detection on the PASCAL VOC2012 dataset (Everingham et al.) with a budget of 3,000 images. Since each image may contain a mixture of objects from different classes, we define the data source as a set of different classes whose objects are likely to appear in the same image, i.e., indoor, wildlife, transport, and human. Then we partition the dataset into four subsets,

each containing a collection of images from one of the four data sources. Details of data source assignment is given in Appendix B. We use mean Average Precision with an intersection-over-union (IOU) threshold of 0.5 (mAP@50) to measure the performance of the algorithms within each subset. The metrics are the mean and minimum mAP@50 among all subsets of images. We use the SSD300 (Liu et al., 2016) architecture with input image size of 300 for this task. The backbone network is VGG16 (Liu & Deng, 2015) pretrained on ImageNet (Deng et al., 2009).

**Vision-Language Modeling.** We perform a simple Visual Question Answering (VQA) task under a budget of 1,000 question-answer pairs from the VQAv2 dataset (Antol et al., 2015) using the proposed algorithms. We partition the dataset into three data sources based on the type of the questions, i.e., yes/no questions, numerical questions, and descriptive questions. The metrics are the mean and minimum per-token accuracies of each data source. We adopt a pretrained SmolVLM-256M-Base model (Marafioti et al., 2025) for this task.

**Single-class Object Detection.** We further implement the proposed algorithms on the COSMOS testbed (Raychaudhuri et al., 2020) for detecting vehicles in an urban intersection (our motivating example in Section 1) with a budget of 2,500 images. The COSMOS testbed includes a traffic intersection in New York City, with three cameras overlooking the traffic flows from different angles. Further details of the testbed setup is given in Appendix B. We collect and annotate the captured images from the three cameras in the testbed and consider each camera as a data source. The metrics are the mean and minimum Average Precision with IOU threshold of 0.5 (AP@50) among all cameras. We use the same model architecture as the one used for the above multi-class task but with an input image size of 320.

To understand how the three algorithms differ from each other and what training configurations to use, we run the classification task on the MNIST dataset under various setups. Each algorithm executes 1,000 rounds and collects a batch of 8 samples every 4 rounds under a total budget of 2,000 training images. The results are depicted in Figure 1. It can be observed that Eps-OGD and UCB-OGD consistently outperform Rand-OGD (Figure 1(a)), with UCB-OGD exhibiting comparatively better accuracy. The colored band associated with each line represents the range of minimum and maximum class-wise accuracy of each algorithm. We also observe from Figure 1(b) that the Adam optimizer with cosine-annealing learning rate scheduler (LRS) and L2 regularization (Reg) provides the smoothest trajectory, which we adopt for the following experiments. Figure 1(c) shows the distribution of the samples collected from each digit. It can be seen that UCB-OGD tends to explore the data sources with fewer samples more aggressively compared to Eps-OGD. Other details of the implementation are given in Appendix B.

Table 1. Performance of the proposed algorithms, UCB-OGD and Eps-OGD, compared to Rand-OGD and active learning algorithms on standard datasets and complex real-world tasks.

| DATASET (BUDGET) | MODEL | ALG | MIN ACC | MEAN ACC |
|---|---|---|---|---|
| CIFAR10 (10K) | CNN | UC | 49.3 | **68.7** |
| | | EN | 53.0 | 68.1 |
| | | BALD | 45.0 | 66.5 |
| | | DBAL | **52.7** | **68.6** |
| | | BADGE | 40.0 | 61.0 |
| | | RAND-OGD | 36.3 | 63.3 |
| | | EPS-OGD | 48.9 | 64.5 |
| | | UCB-OGD | 52.3 | 66.5 |
| VOC2012 (3K) | SSD300 | MDN | 42.2 | 47.2 |
| | | RAND-OGD | 40.6 | 51.3 |
| | | UCB-OGD | **44.7** | **53.0** |
| VQAv2 (1K) | SMOLVLM | RAND-OGD | 20.9 | 20.9 |
| | | UCB-OGD | **22.6** | **22.9** |
| TESTBED (2K5) | SSD300 | RAND-OGD | 57.0 | 66.7 |
| | | UCB-OGD | **61.7** | **69.2** |

We also draw comparisons with several state-of-the-art AL algorithms. For the classification task, we consider several well-known AL algorithms in the literature (Munjal et al., 2022), i.e., Uncertainty-based Sampling (UC), Entropy-based Sampling (EN), Bayesian Active Learning by Disagreement (BALD), Deep Bayesian Active Learning (DBAL), and Deep Batch Active Learning (BADGE) (Ash et al., 2020). All AL algorithms are given an initial *labeled* pool of 1,000 samples (10% of budget), and proceeds to collect 3,000 samples in each episode from the remaining dataset for three episodes. For the multi-class object detection task, we consider the Mixture Density Network (MDN) that takes a probabilistic approach for uncertainty measurement (Choi et al., 2021). The MDN algorithm is given an initial *labeled* pool of 600 samples (20% of budget), and proceeds to collect 800 samples per episode for three episodes. We note that the sizes of the initial labeled pool for both tasks are typically smaller than the common setup in AL literature in order to emulate data-scarce scenarios. The results are summarized in Table 1.

For CIFAR10, it can be seen that while all algorithms outperform the Rand-OGD baseline, DBAL and UCB-OGD give similar minimum accuracies of 52.7 and 52.3, surpassing other algorithms by a noticeable margin. Meanwhile, two AL methods, UC and DBAL, give a higher mean accuracy of 68.7 and 68.6 over all classes.

For VOC2012, UCB-OGD outperforms both Rand-OGD and MDN in both minimum and mean mAP@50. Furthermore, we inspect the effect of the initial pool size on MDN.

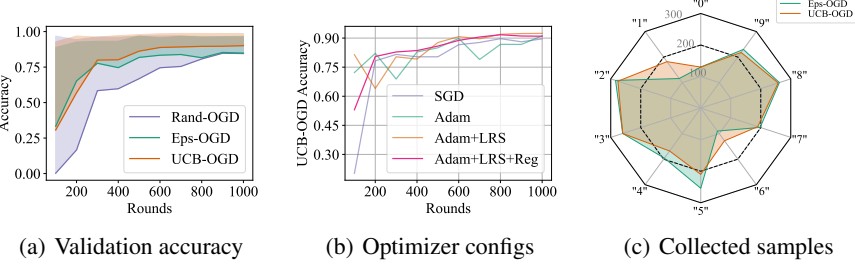

(a) Validation accuracy     (b) Optimizer configs     (c) Collected samples

*Figure 1.* Comparisons of Rand-OGD, Eps-OGD, and UCB-OGD on MNIST over 1,000 rounds in terms of validation accuracy, optimization configurations, and the distribution of the number of samples collected from each data source (i.e., digit).

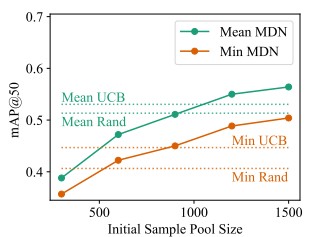

*Figure 2.* Initial MDN pool size *v.s.* the final mAP@50 compared to Rand-OGD and UCB-OGD.

We fix the total budget of 3,000 samples and change the number of samples allocated during each episode. The results are shown in Figure 2. It can be observed that MDN requires 1,200 initially labeled samples (40% of budget) in order to top UCB-OGD on both the minimum and mean mAP@50. This finding affirms that pool-based AL algorithms like MDN can be sensitive to the initial sample pool, and our proposed method is more suitable when the amount of labeled samples is limited.

We further note that AL methods achieve the reported performances by incrementally sampling from the entire unlabeled pool, while our methods only make decisions based on the annotated portion of the dataset, which is much smaller. Appendix C provides other experimental results such as the performances of new models trained on the collected data.

For the VQA task, UCB-OGD outperforms Rand-OGD on both mean and minimum accuracy. Although larger experiments are still required, we believe the results are able to demonstrate that our proposed framework can be applied to complex tasks like vision-language modeling where the training data incorporates various modalities. This is especially beneficial for training small models (such as Smol-VLM) under scenarios where the data budget and computational resources are both limited.

Finally, our experiments on the testbed show that UCB-OGD achieves a 4.7 improvement in minimum AP@50 compared to Rand-OGD. Moreover, while Rand-OGD reports 57.0 AP@50 after collecting 2,500 samples, UCB-OGD achieves the same milestone with only 2,160 samples, saving more than 13% of the total budget.

## 6. Conclusions and Future Work

We introduce an adaptive data collection framework that enables robust learning across multiple distributions under a fixed data collection budget constraint. Our theoretical analysis establishes a general minimax regret guarantee. Moreover, our method consistently outperforms existing base-lines, including random sampling and several well-known Active Learning (AL) approaches. By optimizing data collection decisions, our framework achieves comparable or better model performance with fewer (labeled and unlabeled) samples, effectively reducing annotation costs while improving generalization across heterogeneous distributions in real-world deployment. These results highlight the potential of integrating online optimization and bandit-based sampling for efficient data acquisition, offering a scalable solution for robust learning in real-world applications.

An important direction for future work is to further tighten the regret bounds for more restrictive types of objective functions and verify their applicability in DNNs. Incorporating a Bayesian perspective could further improve the efficiency during sampling. For the experimental verifications, larger-scale evaluations on diverse, real-world datasets with more complex multi-modal distributions would provide deeper insights into the effectiveness of the proposed framework. Expanding experiments to include dynamic environments, such as continuously evolving traffic patterns in smart cities, would further validate the framework's robustness and scalability to broader applications.

## Acknowledgements

This work was supported in part by NSF grant CNS-1827923 and EEC-2133516, NSF grant CNS-2038984 and corresponding support from the Federal Highway Administration (FHA), MediaTek Inc USA, NSF grant CNS-2148128 and by funds from federal agency and industry partners as specified in the Resilient & Intelligent NextG Systems (RINGS) program, ARO grant W911NF2210031, and compute resources from NVIDIA Academic Grant "Edge AI for Equitable and Safe Intersections in Urban Metropolises". We would also like to thank the anonymous reviewers for their suggestions and insights that improved this work.

## Impact Statement

This paper presents work whose goal is to advance the field of Machine Learning. There are many potential societal consequences of our work, none which we feel must be specifically highlighted here.

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

# A. Proofs

## A.1. Proof of Proposition 4.2

We first restate Assumption 2.4 on the domain of $\theta$.

**Assumption A.1** (Finite Domain). The model parameters generated by Algorithm 1 in all rounds, i.e., $\theta_1, \ldots, \theta_T$, satisfies $\mathbb{E}[\|\theta_t - \tilde{\theta}^\star\|] \leq D$ for some $D \geq 0$, where $\tilde{\theta}^\star := \arg\min_\theta \sum_{t=1}^T \mu_{k_t}(\theta)$.

We further state the following lemma (Garrigos & Gower, 2024).

**Lemma A.2** (Gradient Norm). *Let Assumption 2.3 and Assumption A.1 hold. Then for some set of inputs $\mathcal{S}$, we have*

$$\mathbb{E}[\|\nabla\hat{\mu}_k(\theta; \mathcal{S}) - \nabla\mu_k(\theta)\|^2] \leq \frac{2(L^2 + \sigma^2)}{|\mathcal{S}_k|}.$$

Now we prove Theorem 3.1.

*Proof.* By the property of convex functions,

$$\mu_{k_t}(\theta_t) - \mu_{k_t}(\tilde{\theta}^\star) \leq \langle \nabla\mu_{k_t}(\theta_t), \theta_t - \tilde{\theta}^\star \rangle,$$

where $\tilde{\theta}^\star$ is defined in Assumption A.1 and $\langle x, y \rangle$ is the inner product between vectors $x, y$ in $\Theta$. Since $\theta_{t+1} = \theta_t - \eta_t \nabla\hat{\mu}_{k_t}(\theta_t; \cdot)$, we have

$$\|\theta_{t+1} - \theta^\star\|^2 = \|\theta_t - \theta^\star\|^2 + \eta_t^2 \|\nabla\hat{\mu}_{k_t}(\theta_t; \cdot)\|^2 - 2\eta_t \langle \nabla\hat{\mu}_{k_t}(\theta_t; \cdot), \theta_t - \tilde{\theta}^\star \rangle.$$

By rearranging and taking the expectation, we have

$$2\mathbb{E}[\langle \nabla\tilde{\mu}_{k_t}(\theta_t), \theta_t - \tilde{\theta}^\star \rangle] = \mathbb{E}[\mathbb{E}[2\langle \nabla\hat{\mu}_{k_t}(\theta_t; \cdot), \theta_t - \tilde{\theta}^\star \rangle | \mathcal{A}_T]]$$
$$= \eta_t^{-1}\mathbb{E}[\|\theta_t - \theta^*\|^2] - \eta_i^{-1}\mathbb{E}[\|\theta_{t+1} - \tilde{\theta}^\star\|^2] + \eta_t \mathbb{E}[\|\nabla\hat{\mu}_{k_t}(\theta_t; \cdot)\|^2]'$$

Further let $\eta_0^{-1} := 0$. Then the optimization regret in Equation (11) writes

$$2\mathbb{E}[R_o(\mathcal{A}_T)] = 2\sum_{t=1}^T \mathbb{E}[\tilde{\mu}_{k_t}(\theta_t) - \tilde{\mu}_{k_t}(\tilde{\theta}^\star)]$$
$$\leq 2\sum_{t=1}^T \mathbb{E}[\langle \tilde{\mu}_{k_t}(\theta_t), \theta_i - \tilde{\theta}^\star \rangle]$$
$$\leq \sum_{t=1}^T \left( \eta_t^{-1}\mathbb{E}[\|\theta_t - \tilde{\theta}^\star\|^2] - \eta_t^{-1}\mathbb{E}[\|\theta_{t+1} - \tilde{\theta}^*\|^2] + \eta_t \mathbb{E}[\|\nabla\hat{\mu}_{k_t}(\theta_t; \cdot)\|^2] \right)'$$
$$\leq D^2 \sum_{t=1}^T (\eta_t^{-1} - \eta_{t-1}^{-1}) + 2(L^2 + \sigma^2) \sum_{t=1}^T \eta_t$$

where the inequality is a result of Assumption A.1 and Lemma A.2 with $|\mathcal{S}_k| \geq 1$. Setting $\eta_t = 1/(2L\sqrt{t})$ for $t \geq 1$ and recalling that $\sum_{t=1}^T 1/\sqrt{t} \leq 2\sqrt{T}$, we have

$$2\mathbb{E}[R_o(\mathcal{A}_T)] \leq 2L(D^2 + 1)\sqrt{T} + 2L^{-1}\sigma^2\sqrt{T}.$$

This concludes the proof. □

## A.2. Proof of Lemma 4.3 and Lemma 4.4

For Lemma 4.3, we construct the martingale as follows.

*Proof.* For each $X_i \in \mathcal{S} \subseteq \mathcal{X}_k$ and some $k$, let

$$Z_i := \ell(\theta_t, X_i) - \tilde{\mu}_k(\theta_t), \quad i = 1, \ldots, |\mathcal{S}_k|.$$

It is easy to verify that the sequence $\{Z_i, i = 1, \ldots, |\mathcal{S}_k|\}$ is a *martingale difference sequence*, i.e.,

$$\mathbb{E}[Z_i | \mathcal{A}_t] = \mathbb{E}[\ell(\theta_t, X_i) | \mathcal{A}_t] - \tilde{\mu}_k(\theta_t) = 0.$$

Applying the Azuma's inequality on $\{Z_i, i = 1, \ldots, |\mathcal{S}_k|\}$ gives the statement of the lemma. □

To prove Lemma 4.4, simply note that any confidence radius $r_k(\cdot)$ in round $t$ is *adapted to* the algorithm decisions $\mathcal{A}_t$ and apply Lemma 4.3.

*Proof.* We start by writing

$$\begin{aligned}
\Delta_k(\theta_t) &= \tilde{\mu}^\star(\theta_t) - \tilde{\mu}_k(\theta_t) \\
&= \left(\hat{\mu}_k(\theta_t; \mathcal{S}) - \tilde{\mu}_k(\theta_t)\right) - \left(\hat{\mu}^\star(\theta_t; \mathcal{S}) - \tilde{\mu}^\star(\theta_t)\right) + \hat{\Delta}_k(\theta_t; \mathcal{S})^.
\end{aligned}$$

For the term $\hat{\mu}_k(\theta_t; \mathcal{S}) - \tilde{\mu}_k(\theta_t)$, denote

$$\delta_t := \Pr\left[\hat{\mu}_k(\theta_t; \mathcal{S}) - \tilde{\mu}_k(\theta_t) \geq r_k(\mathcal{S}) \big| \mathcal{A}_t\right].$$

Taking its expectation conditioned on $\mathcal{A}_t$ and $\mathcal{S}$, we have

$$\hat{\mu}_k(\theta_t; \mathcal{S}) - \mathbb{E}[\tilde{\mu}_k(\theta_t) | \mathcal{A}_t, \mathcal{S}] \leq (1 - \delta_t) r_k(\mathcal{S}) + C\delta_t \leq r_k(\mathcal{S}) + C\delta_t.$$

And similarly for $\hat{\mu}^\star(\theta_t; \mathcal{S}) - \tilde{\mu}^\star(\theta_t)$,

$$\hat{\mu}^\star(\theta_t; \mathcal{S}) - \mathbb{E}[\tilde{\mu}^\star(\theta_t) | \mathcal{A}_t, \mathcal{S}] \geq -(1 - \delta_t) r^\star(\mathcal{S}) - C\delta_t \geq -r^\star(\mathcal{S}) - C\delta_t.$$

Plugging back into the first equation in the proof gives the result of the lemma, where the value of $\delta_t$ can be obtained by applying the definition of the confidence radius in Equations (7) and (17) to Lemma 4.3, which are of order $O(t^{-\alpha})$. □

### A.3. Proof of Proposition 4.5

First we prove the bandit regret for Eps-OGD, which is adapted from the proof of the $\epsilon$-Greedy algorithm for stochastic bandits (Auer et al., 2002).

*Proof.* Since a greedy step picks $k_t$ to be $\arg\max_k \hat{\mu}_k(\theta_t | \mathcal{X}_{t-1})$, we have $\hat{\mu}_{k_t}(\theta_t | \mathcal{X}_{t-1}) \geq \hat{\mu}^\star(\theta_t | \mathcal{X}_{t-1})$. This implies that we observe $\hat{\Delta}_{k_t}(\theta_t | \mathcal{X}_{t-1}) = 0$ with probability $1 - \epsilon_t$. Otherwise, an $\epsilon$-exploration step picks $k_t \sim U(\{1, \ldots, K\})$ and $\Delta_{k_t}(\theta_t | \mathcal{X}_{t-1}) \leq C$ (Assumption 2.3). More formally, when $t = 1$, we set $\epsilon_t = 1$ and the data source is selected randomly with $\mathbb{E}[\Delta_{k_t}(\theta_t)] \leq C$. For $t > 1$, we can write

$$\mathbb{E}[\hat{\Delta}_{k_t}(\theta_t; \mathcal{X}_{t-1})] \leq C\epsilon_t.$$

According to Lemma 4.4, we have

$$\mathbb{E}\left[\Delta_{k_t}(\theta_t) \big| \mathcal{A}_t, \mathcal{X}_{t-1}\right] \leq r_k(\mathcal{X}_{t-1}) + r^\star(\mathcal{X}_{t-1}) + \hat{\Delta}_{k_t}(\theta; \mathcal{X}_{t-1}) + 2C\delta_t.$$

Combining the two equations gives

$$\mathbb{E}[\Delta_{k_t}(\theta_t) | \mathcal{A}_t] \leq r_k(\mathcal{X}_{t-1}) + r^\star(\mathcal{X}_{t-1}) + 2C\delta_t + C\epsilon_t.$$

For the Eps-OGD algorithm, the confidence radius defined in Equation (17) writes

$$r_k(\mathcal{X}_{t-1}) := C\sqrt{\frac{\alpha K \ln t}{2|\mathcal{X}_{t-1}|\epsilon_t}} = C\sqrt{\frac{\alpha K \ln t}{2M(t-1)\epsilon_t}}, \quad t > 1.$$

Then the concentration probability given by Lemma 4.3 is $\delta_t = t^{-\alpha \frac{K|\mathcal{X}_{t-1,k}|}{M(t-1)\epsilon_t}}$. Assume that $\epsilon_t$ decreases monotonically with $t$. Then $\mathbb{E}[\|\mathcal{X}_{t-1,k}\|] \geq M(t-1)\epsilon_t/K$ for all $k$, yielding $\mathbb{E}[\delta_t] \leq t^{-\alpha}$. Thus,

$$\mathbb{E}[\Delta_{k_t}(\theta_t)] \leq 2C\sqrt{\frac{\alpha K \ln t}{2M(t-1)\epsilon_t}} + 2Ct^{-\alpha} + B\epsilon_t.$$

This is minimized by taking $\epsilon_t = \sqrt[3]{\alpha K \ln t/(2M(t-1))}/2$ as in Equation (5), which gives

$$\mathbb{E}[\Delta_{k_t}(\theta_t)] \leq (2\sqrt{2}+1)C\sqrt[3]{\frac{\alpha K \ln t}{2M(t-1)}} + 2Ct^{-\alpha}.$$

To compute the total expected regret, we first bound the following summation

$$\sum_{t=2}^{T} \sqrt[3]{\frac{\ln t}{t-1}} \leq \sqrt[3]{\ln T} \cdot \sum_{t=2}^{T} \frac{1}{\sqrt[3]{t-1}}$$

$$\leq \sqrt[3]{\ln T} \int_0^T x^{-\frac{1}{3}} dx$$

$$= \frac{3}{2}\sqrt[3]{T^2 \ln T}$$

For any $\alpha \geq 1/2$, we have $\sum_{t=1}^{T} \delta_t \leq 2\sqrt{T}$. Thus, we can write

$$\mathbb{E}[R_b(\mathcal{A}_T)] = C + \sum_{t=2}^{T} \mathbb{E}[\Delta_{k_t}(\theta_t)] \leq C + \frac{3(2\sqrt{2}+1)C}{2}\sqrt[3]{\frac{\alpha KT^2 \ln T}{2M}} + 4C\sqrt{T}.$$

This concludes the proof. $\square$

Now we prove the bandit regret of UCB-OGD.

*Proof.* Since UCB selection picks $k_t$ to be $\arg\max_k \hat{\mu}_k(\theta_t|\mathcal{X}_{t-1}) + r_k(\mathcal{X}_{t-1})$, we have

$$\hat{\mu}^\star(\theta_t|\mathcal{X}_{t-1}) + r^\star(\mathcal{X}_{t-1}) \leq \hat{\mu}_{k_t}(\theta_t|\mathcal{X}_{t-1}) + r_{k_t}(\mathcal{X}_{t-1}),$$

or, equivalently,

$$\hat{\Delta}_{k_t}(\theta_t|\mathcal{X}_{t-1}) \leq r_{k_t}(\mathcal{X}_{t-1}) - r^\star(\mathcal{X}_{t-1}).$$

Combining with Lemma 4.4, we have

$$\mathbb{E}[\Delta_{k_t}(\theta_t)|\mathcal{A}_t, \mathcal{X}_{t-1}] \leq 2r_{k_t}(\mathcal{X}_{t-1}) + 2C\delta_t.$$

Recall the definition of confidence radius $r_k(\mathcal{X}_{t-1})$ in Equation (7), with $\delta_t \leq t^{-\alpha}$ given by Lemma 4.3. Further let $n_k(t)$ denote the number of times a data source $k$ is selected up to time $t$. We can write $|\mathcal{X}_{t,k}| = Mn_k(t)$ for any $k, t$. Then the equation above can be written as

$$\mathbb{E}[\Delta_{k_t}(\theta_t)] \leq 2C\sqrt{\frac{\alpha \ln t}{2M}}\mathbb{E}\left[\sqrt{\frac{1}{n_k(t-1)}}\right] + 2C\delta_t.$$

Let $\mathcal{A}_{T,k} := \{(k_t, \theta_t) \in \mathcal{A}_T : k_t = k\}$. We assume that the algorithm iterates over every data source during the first $K$

rounds for initial warm-up. Then, the regret incurred by some arm $k$ can be written as

$$
\begin{aligned}
\mathbb{E}[R_b(\mathcal{A}_{T,k})] &= \sum_{t=1}^{T} \mathbb{E}\left[\mathbb{1}\left\{k_t = k\right\} \Delta_{k_t}(\theta_t)\right] \\
&\leq C + \sum_{t=K+1}^{T} \mathbb{E}\left[\mathbb{1}\left\{k_t = k\right\} \Delta_{k_t}(\theta_t)\right] \\
&\leq C + \sum_{t=K+1}^{T} \left(2C\sqrt{\frac{\alpha \ln t}{2M}} \mathbb{E}\left[\frac{\mathbb{1}\left\{k_t = k\right\}}{\sqrt{n_k(t-1)}}\right] + 2C\delta_t\right) \\
&\leq C + C\sqrt{\frac{2\alpha \ln T}{M}} \mathbb{E}\left[\sum_{t=K+1}^{T} \frac{\mathbb{1}\left\{k_t = k\right\}}{\sqrt{n_k(t-1)}}\right] + 2C\mathbb{E}\left[\sum_{t=K+1}^{T} \delta_t \mathbb{1}\left\{k_t = k\right\}\right]
\end{aligned}
$$

Notice that between any consecutive rounds $t-1$ and $t$, $n_k(t)$ is increased by 1 *if and only if* $k_t = k$ (i.e., the data source is selected and new samples are added), or the numerator is zero otherwise, hence,

$$
\begin{aligned}
\sum_{t=K+1}^{T} \frac{\mathbb{1}\left\{k_t = k\right\}}{\sqrt{n_k(t-1)}} &= \frac{1}{1} + \frac{0}{1} + \cdots + \frac{0}{1} + \frac{1}{\sqrt{2}} + \frac{0}{\sqrt{2}} + \cdots \\
&\quad + \cdots + \frac{0}{\sqrt{n_k(T-2)}} + \frac{1}{\sqrt{n_k(T-1)}} \,. \\
&= \sum_{n=1}^{n_k(T-1)} \frac{1}{\sqrt{n}} \\
&\leq 2\sqrt{n_k(T-1)}
\end{aligned}
$$

Then expected regret can be written as

$$
\begin{aligned}
\mathbb{E}[R_b(\mathcal{A}_T)] &= \sum_{k=1}^{K} \mathbb{E}[R_b(\mathcal{A}_{T,k})] \\
&\leq KC + 2C\sqrt{\frac{2\alpha \ln T}{M}} \mathbb{E}\left[\sum_{k=1}^{K} \sqrt{n_k(T-1)}\right] + 2C\mathbb{E}\left[\sum_{k=1}^{K}\sum_{i=1}^{N} \delta_t \mathbb{1}\left\{k_t = k\right\}\right], \\
&\leq KC + 2C\sqrt{\frac{2\alpha \ln T}{M}} \mathbb{E}\left[\sqrt{K\sum_{k=1}^{K} n_k(T-1)}\right] + 2C\mathbb{E}\left[\sum_{t=1}^{T} \delta_t\right]
\end{aligned}
$$

where the last line follows from the inequality between arithmetic mean and quadratic mean, and the fact that $\sum_{k=1}^{K} \mathbb{1}\left\{k_t = k\right\} = 1$ for all $t$. Further recall that $\sum_{k=1}^{K} n_k(t) = t$. Then, for any $\alpha \geq 1/2$, we have

$$
\mathbb{E}[R_b(\mathcal{A}_T)] \leq KC + 2C\sqrt{\frac{2\alpha KT \ln T}{M}} + 4C\sqrt{T}.
$$

This concludes the proof. □

### A.4. Mean Convergence

For the *Rand-OGD* algorithm, since it enforces that the data sources are queried in a balanced way, the algorithm reduces to a standard stochastic gradient descent for minimizing the average loss over the data sources, i.e., $\sum_{k=1}^{K} \mu_k(\theta)/K$. We have the following result.

**Theorem A.3.** *Let $\mu_1, \ldots, \mu_K$ be convex in $\theta$. The Rand-OGD algorithm satisfies*

$$
\mathbb{E}\left[\frac{1}{K}\sum_{k=1}^{K} \mu_k(\bar{\theta}_{\mathcal{A}_T}) - \min_\theta \frac{1}{K}\sum_{k=1}^{K} \mu_k(\theta)\right] \leq O(T^{-\frac{1}{2}}). \tag{22}
$$

*Proof.* Denote $\bar{\mu}(\theta) := \sum_{k=1}^{K} \mu_k(\theta)/K$. Since $k_t \sim \mathrm{U}(\{1, \ldots, K\})$, it holds for any fixed $\theta$ that $\mathbb{E}[\mu_{k_t}(\theta)] = \bar{\mu}(\theta)$. By the definition of the optimization regret (Equation (11)) and the generalization regret (Equation (13), defined on a single source $\bar{\mu}$),

$$
\begin{aligned}
\mathbb{E}\left[\sum_{t=1}^{T} \bar{\mu}(\theta_t) - T \min_{\theta} \bar{\mu}(\theta)\right] &= \mathbb{E}\left[\sum_{t=1}^{T} \mathbb{E}[\mu_{k_t}(\theta_t)|\theta_t] - T \min_{\theta} \bar{\mu}(\theta)\right] \\
&= \sum_{t=1}^{T} \mathbb{E}[\mu_{k_t}(\theta_t)] - \min_{\theta} \sum_{t=1}^{T} \mathbb{E}[\mu_{k_t}(\theta)] \\
&\leq \sum_{t=1}^{T} \mathbb{E}[\mu_{k_t}(\theta_t) - \tilde{\mu}_{k_t}(\theta_t)] + \sum_{t=1}^{T} \mathbb{E}[\tilde{\mu}_{k_t}(\theta_t)] - \min_{\theta} \sum_{t=1}^{T} \mathbb{E}[\tilde{\mu}_{k_t}(\theta)] \\
&= \mathbb{E}[R_g(\mathcal{A}_T) + R_o(\mathcal{A}_T)]
\end{aligned}
$$,

where the inequality follows from Equation (10). By the convexity of $\mu_k$ and Jensen's inequality, we have $\bar{\mu}(\bar{\theta}_{\mathcal{A}_T}) \leq \sum_{t=1}^{T} \bar{\mu}(\theta_t)/T$. Recalling the upper bound of optimization regret in Proposition 4.2 and the generalization regret in Equation (21) concludes the proof. $\qquad\square$

## A.5. Proof of Pareto-Stationarity

**Theorem A.4** (Pareto Staionarity). *Let $\mathcal{A}$ be a time-smoothed OGD-based algorithm. Then for some $t \sim \mathrm{U}(\{1, \ldots, T\})$, $\theta_t$ is (asymptotically) Pareto-stationary for $\mu_1, \ldots, \mu_K$ as $T, w \to \infty$.*

*Proof.* In this proof, we adopt the time-smoothed OGD framework (Hazan et al., 2017). The time-smoothed gradient at round $t$ w.r.t. some window $w \in [1, T]$ is defined as

$$
\nabla \bar{\mu}_k^w(\theta) := \frac{1}{w} \sum_{i=1}^{w-1} \nabla \mu_{k_{t-i}}(\theta_t).
$$

All $\mu_{k_t}$ where $t \leq 0$ are set to 0 for uniformity. The $w$-local regret is defined as

$$
R_l^w(\mathcal{A}_T) := \sum_{t=1}^{T} \left\|\nabla \bar{\mu}_{k_t}^w(\theta_t)\right\|^2.
$$

A proper time-smoothed OGD algorithm $\mathcal{A}_T$ incurs a $w$-local regret of order $O(T/w^2)$ (Hazan et al., 2017; Hallak et al., 2021). Further, the relationship between individual $\nabla \bar{\mu}_{k_t}^w(\theta_t)$ and the local regret can be given by

$$
\mathbb{E}_{t \sim \mathrm{U}(\{1,\ldots,T\})}\left[\left\|\nabla \bar{\mu}_{k_t}^w(\theta_t)\right\|^2\right] \leq \frac{\mathbb{E}[R_l^w(\mathcal{A}_T)]}{T} = O(1/w^2).
$$

Let $n_k^w(t) := \sum_{i=1}^{w-1} \mathbb{1}\{k_{t-i} = k\}$ be the number of times an arm $k$ is selected from round $t - w + 1$ to round $t$. Then we can rewrite

$$
\nabla \bar{\mu}_{k_t}^w(\theta_t) = \sum_{k=1}^{K} \frac{n_k^w(t)}{w} \nabla \mu_k(\theta_t).
$$

It follows that

$$
E_{t \sim \mathrm{U}(\{1,\ldots,T\})}\left[\left\|\sum_{k=1}^{K} \frac{n_k^w(t)}{w} \nabla \mu_{k_t}(\theta_t)\right\|^2\right] \leq O(1/w^2).
$$

Note that $\theta_s$ is called Pareto-stationary (Sener & Koltun, 2018) if there exists $\alpha_1, \ldots, \alpha_K \in \mathbb{R}$ s.t.

$$
\sum_{k=1}^{K} \alpha_k \nabla \mu_k(\theta) = 0, \quad \text{where} \quad \sum_{k=1}^{K} \alpha_k = 1, \ \alpha_k \geq 0.
$$

Indeed, this is the case for our problem by setting $\alpha_k = n_k^w(t)/w$ and taking the limit for both $T$ and $w$. $\qquad\square$

### A.6. Proof of Proposition 3.3

*Proof.* Consider a problem with $K = 1$ data source. Thus the problem reduces to a standard single-objective stochastic optimization problem. Denote the objective function as $\mu(\theta)$. Then for any gradient-based algorithm $\mathcal{A}$ after $T$ rounds, we have

$$
\begin{aligned}
R(\mathcal{A}_T) &= \sum_{t=1}^{T} \mu(\theta_t) - T \min_{\theta} \mu(\theta) \\
&\geq T \left( \mu(\bar{\theta}_{\mathcal{A}_T}) - \min_{\theta} \mu(\theta) \right),
\end{aligned}
\tag{23}
$$

where we recall Jensen's inequality. Note that $\mu(\bar{\theta}_{\mathcal{A}_T}) - \min_{\theta} \mu(\theta)$ is the optimality gap of algorithm $\mathcal{A}$. Let $\mu(\theta)$ be $L$-Lipschitz (Assumption 2.3) but non-smooth (e.g. a DNN with ReLU activations). The optimality gap of gradient methods in general is lower bounded by $O(T^{-\frac{1}{2}})$ (Shamir & Zhang, 2013). Thus the minimax regret is at least of order $O(T^{\frac{1}{2}})$. $\square$

# B. Implementation Details

## B.1. The COSMOS Testbed

The COSMOS (Cloud-enhanced Open Software-defined MObile wireless testbed for city-Scale deployment) testbed (Raychaudhuri et al., 2020), part of the NSF PAWR program, is being deployed in West Harlem in New York City. It supports research on ultra-high bandwidth and ultra-low latency wireless technologies in real-world urban environments. The testbed features programmable infrastructure across multiple layers, including software-defined radios, 28 GHz mmWave modules, optical transport, edge/core cloud components, and comprehensive control software. Its phased urban deployment enables diverse experimental research and serves as a valuable educational platform.

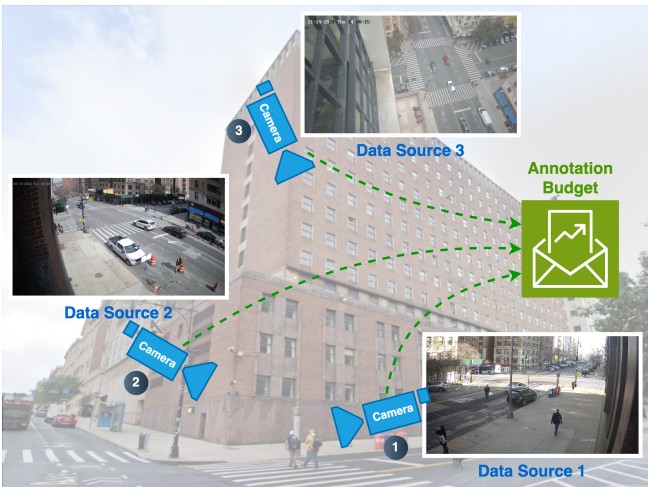

*Figure 3.* Camera setup of an intersection in the COSMOS Testbed.

The testbed includes multiple cameras deployed on the exterior of a multi-story building, illustrated in Figure 3. Each of the cameras provides a distinct viewpoint of the traffic flow, three of which were used as separate data sources for data collection in this paper. These strategically positioned cameras enable comprehensive coverage for urban object detection, facilitating cross-view analysis of vehicles, pedestrians, and street activity.

## B.2. Data Collection Schedule

- **CIFAR10:** We execute our algorithms for 20,000 rounds and collect a batch of 32 samples every 60 rounds until reaching the budget.

- **PASCAL VOC2012:** We execute our algorithms by pretraining for 10,000 rounds (freezing the backbone), collecting a batch of 8 samples every 50 rounds. Then we finetune for 20,000 rounds, collecting a batch of 8 samples 100 rounds until reaching the budget.

- **Testbed:** We execute our algorithms by pretraining for 10,000 rounds (freezing the backbone), collecting a batch of 8 samples every 50 rounds. Then we finetune for 20,000 rounds, collecting a batch of 8 samples 200 rounds until reaching the budget.

## B.3. Data Processing for VOC2012

We partition the images in PASCAL VOC2012 data set into the following data sources based on their concurrence within the same image:

- **Indoor:** cat, dog, TV monitor, sofa, bottle, potted plant, chair, and dining table.

- **Transport:** aeroplane, train, boat, motorbike, bicycle, car, and bus.

- **Wildlife:** bird, horse, cow, and sheep.

- **Human:** person only.

An image is partitioned to the *human* data source only if no other classes appear. If objects of multiple sources appear in the same image, then the upper one in the list above takes priority (e.g. if an image contains both a sofa and a bird, then it is partitioned to the *indoor* data source).

Further, for every data source, a class is removed from mAP calculation if has less than 20 objects in all the images from that source in the validation set.

## C. Generalization Results

We inspect the generalization ability of the proposed data collection framework by training other models using classical training loops on the data collected by uniform allocation strategy and the UCB-OGD strategy. The chosen model is SSD300 for the PASCAL VOC2012 dataset and YOLOv8 for the testbed dataset.

*Table 2.* Comparisons of the performance of new models trained on the data collected by different strategies.

| DATA SOURCE (BUDGET) | MODEL | ALLOCATION | MIN ACC | MEAN ACC |
|---|---|---|---|---|
| VOC2012 (3K) | SSD300 | UNIFORM | 35.0 | 51.7 |
| | | UCB | **36.2** | **52.2** |
| TESTBED (2K5) | YOLOv8 | UNIFORM | 62.2 | 72.5 |
| | | UCB | **63.7** | **72.9** |

