# OpenReview forum: "Adaptive Data Collection for Robust Learning Across Multiple Distributions"
_ICML.cc/2025/Conference — ICML 2025 poster_

### Official Review · Reviewer_WqSu · 2025-02-17

**Overall Recommendation:** 3

**Summary:**

This paper considers a multi-round decentralized training method with a limited annotation budget to label data from multiple distributions from various locations.

**Claims And Evidence:**

Yes, the claims have theoretical proofs besides empirical experiments.

**Essential References Not Discussed:**

In other application domains, there's a thread on decentralized data collection with theoretical guarantees compared to the optimality gap. Some use optimization-based methods, and some use submodular methods, which have pretty straight-forward proofs on the optimality gap. Please compare them in the paper and add them as baselines.


[1] When data acquisition meets data analytics: A distributed active learning framework for optimal budgeted mobile crowdsensing. Qiang Xu; Rong Zheng, INFOCOM 2017
[2] Decentralized Data Collection for Robotic Fleet Learning: A Game-Theoretic Approach. Oguzhan Akcin, Po-han Li, Shubhankar Agarwal, Sandeep P. Chinchali, CoRL 2022
[3] Fleet Active Learning: A Submodular Maximization Approach. Oguzhan Akcin, Orhan Unuvar, Onat Ure, Sandeep P. Chinchali, CoRL 2023
[4] Distributed Submodular Maximization. Baharan Mirzasoleiman, Amin Karbasi, Rik Sarkar, Andreas Krause. JMLR 2016

**Experimental Designs Or Analyses:**

Yes, I checked all the results and raised my question in the "questions" section.

**Methods And Evaluation Criteria:**

Yes, the paper shows results in multiple domains, such as classification tasks.

**Other Comments Or Suggestions:**

N/A.

**Other Strengths And Weaknesses:**

# Strengths
1. The proposed method can be combined with other existing bandit algorithms.
2. The paper clearly stated the assumptions and the claims.

# Weaknesses
1. The IID assumption (Assumption 2.6) is not so realistic since in the real world, the data distributions are correlated if the locations are close enough (following the example in the intro). Please justify.
2. The paper mainly leverages existing results on bandits to their data collection problem.

**Questions For Authors:**

1. In Table. 1, UCB-OGD is not actually on par with DBAL since the MEAN ACC of DBAL is much better. Then, what is the point of proposing UCB-OGD? Maybe try them on a more complicated dataset (ImageNet) to demonstrate their capability.
2. The definition of data sources is strongly related to the core definition of a task. In the TestBed experiment, the cameras are correlated in some ways we do not even know and the algorithms still work. It thus adds complexity to the setting and worth investigating. Can you do the same with the other two experiments such that the objects appear in the data sources are correlated, and conduct performance analysis on them? Also, it may reflect the assumptions made earlier.

**Relation To Broader Scientific Literature:**

This paper is an add-on to existing bandit and decentralized data collection problems.

**Theoretical Claims:**

The paper mainly leverages existing results from bandit algorithms.

---

> ### Author Rebuttal · Authors · 2025-04-01
>
> Dear reviewer WqSu,
>
> Thank you for your questions. We would like to further clarify the problem and our contributions.
>
> **Problem Definition and Connection to Decentralized Data Collection:** The adaptive data collection problem is indeed related to many topics in active learning (AL), including the suggested line of work on decentralized data collection. Particularly, the theoretical guarantees of decentralized data collection are often achieved by leveraging *predefined* data quality metrics, such as information functions [1], submodular functions [2,4], or a target data distribution [3]. Many centralized AL algorithms (e.g. uncertainty- or entropy-based sampling) share similar intuitions by selecting the most relevant data to annotate using such metrics, which we compare as baselines in *Section 5*. However, the robustness of real-world applications are often measured in terms of their worst-case performances rather than the quality of the data itself. Therefore, it is more common in robust learning to directly minimize the worst-case loss, which is the *minimax objective* employed by our paper. We further discuss the advantages of the proposed framework in the following paragraph. We will also make sure to add the suggested references to the related work of our paper.
>
> **Advantage of The Proposed Framework (question 1):** One of the major advantages of the proposed framework is its high flexibility to apply to a wide range of different tasks without relying on predefined data quality metrics (as in decentralized data collection). This is especially useful in data-scarce scenarios where a target dataset is absent, or in complex tasks (e.g. object detection or VLM training) where the data quality function is hard to define. Although numerous AL algorithms exist for specific tasks such as classification and detection, to the best of our knowledge, there isn't a general framework of adaptive data collection with *provable minimax regret bounds* that incorporates DL models. In fact, we demonstrated in the experiments (*Section 5*) that the same algorithmic framework of UCB-OGD can be utilized to train robust DL models for classification, object detection, and vision-language modeling (VLM) for which we have provided experimental result in *[Figure 5](https://anonymous.4open.science/r/icml2025-adaptive4robust-CB0F/fig5-alcurve-vqav2.pdf)*. We kindly refer to our response to reviewer 7Pps for further descriptions of these results. Therefore, although UCB-OGD shows lower mean accuracy than AL on the CIFAR10 dataset, it achieves similar minimum accuracy (which is our main performance metric), while also yielding superior performances in object detection and VLM tasks (*Table 1* and new *[Figure 4](https://anonymous.4open.science/r/icml2025-adaptive4robust-CB0F/fig4-alcurve-voc2012.pdf)*).
>
> ***I.I.D.* Assumption (weakness 1, question 2):** In *Assumption 2.6*, independence is only assumed *within* each data source, i.e. we assume that the images collected from a single camera are *i.i.d.*, while no assumptions are made with respect to the relationships between the data distributions of different cameras. This is the standard assumption of bandit problems, where the per-arm rewards are required to be *i.i.d.*, but different reward distributions can be arbitrary. In the smart city intersection example, *Assumption 2.6* states that the images from different cameras can be correlated *spatially* (when the locations are close) but not *temporally*. This is the case for the testbed dataset, since the images are sporadically collected during an extended period of time with minimal temporal dependence. We also note that the data sources defined in the VOC2012 experiment indeed admit correlations due to overlapping object classes (details in *Section 5*).
>
> **Our Novelty and Contributions (weakness 2):** While OGD and UCB are both established algorithms, we believe the combination of these techniques for the studied problem is novel. We want to emphasize that UCB has been proposed for *stationary stochastic bandits*, while our problem is a *non-stationary contextual bandit* where vanilla UCB is not applicable (details in *Section 4*). In this paper, we prove that, by maintaining an adaptively collected dataset and leveraging its information, UCB (and other stationary bandit algorithms like $\epsilon$-Greedy) can be combined with online optimization algorithms such as OGD to provide *provable minimax regret bounds* despite the non-stationary nature of our problem. We further note that the proposed framework can be incorporated with various bandit and optimization algorithms tailored to different problems at hand. We believe that our paper shows that adaptive data collection with robust theoretical guarantees can be a promising approach applicable to a wide range of tasks.
>
> We appreciate your suggestions and hope that the answers above could address your concerns. We welcome any further questions or feedback.

---

> > ### Comment · Reviewer_WqSu · 2025-04-01
> >
> > Thanks for the reply. I have increased the score to reflect the additional experiments and clarifications of the authors.

---

### Official Review · Reviewer_p2BW · 2025-03-07

**Overall Recommendation:** 3

**Summary:**

This paper proposes a framework for adaptive data collection aimed at robust learning in multi-distribution scenarios under a fixed data collection budget. The proposed algorithm dynamically selects a data source for sampling in each round, updates the model parameters using gradient descent, and repeats the process to minimize the expected loss across all data sources.

While the paper addresses an interesting problem and provides theoretical guarantees, the lack of significant technical contribution, unclear motivations, and questionable performance metric design weaken its overall impact. The authors should provide stronger justification for their approach, clarify the baseline comparisons, and consider tailoring their algorithm to better address the specific challenges of adaptive data collection in multi-distribution scenarios.

**Claims And Evidence:**

Yes

**Essential References Not Discussed:**

No

**Experimental Designs Or Analyses:**

No

**Methods And Evaluation Criteria:**

Yes

**Other Comments Or Suggestions:**

See my questions.

**Other Strengths And Weaknesses:**

Strengths:

This paper tackles an interesting problem and provides theoretical guarantees.

Weaknesses:
- The proposed method appears to be a direct application of existing approaches—minimizing the maximum loss via UCB sampling combined with OGD. This makes the novelty and technical contribution of the paper seem limited, as no significant algorithmic innovation or optimization is introduced.

- The paper does not clearly articulate why adaptive data collection is necessary or advantageous in this specific context. The problem studied seems to deviate from the original objective of minimizing the expected loss across multiple distributions.

- The algorithm relies on standard stochastic gradient descent without any specific optimization tailored to the adaptive data collection problem in multi-distribution scenarios. This further limits the novelty of the proposed approach.

**Questions For Authors:**

1. What is the motivation for considering adaptive data collection in this specific setting? How does this approach improve over simpler methods that do not rely on dynamic sample selection?

2. The regret is defined based on the $T\min_\theta \max_k \mu_k(\theta)$. However, if we do not employ dynamic sample selection. The baseline should be $T\min_\theta \frac{1}{K}\sum_k \mu_k(\theta)$. It seems that comparing with this baseline can better prove the advantage of your adaptive data allocation.

3. Uniform sampling from all distributions seems to overlook their varying importance. How does the method take into account the relative significance of different distributions in the overall objective function?

4. The algorithm appears to be a straightforward integration of UCB and OGD. Are there any specific innovations or adaptations made to optimize the method for robust learning across multiple distributions?

**Relation To Broader Scientific Literature:**

The paper tackles an important and practical problem—adaptive data collection for robust learning across multiple distributions. This topic is relevant to applications in domains where data is collected from diverse sources with limited budgets.

**Theoretical Claims:**

No

---

> ### Author Rebuttal · Authors · 2025-04-01
>
> Dear reviewer p2BW,
>
> Thank you for your questions. We would like to further clarify the intuitions of the problem and our contributions.
>
> **Necessity of Adaptive Data Collection (Q1):** In our motivating example of vehicle detection in a smart city intersection (*Section 1*), DL models such as SSD and YOLO require a large amount of annotated data for training. However, annotation for complex tasks such as object detection can be quite expensive, and is often subject to budget or resource constraints in a real-world application like ours. Therefore, it is essential to understand how the limited data budget can be utilized *efficiently* to maximize the performance. This is also the core idea of active learning (AL), which we also add as baselines for experimental comparisons. The improvement of the proposed algorithms (Eps-OGD and UCB-OGD) over static (or uniform) sample selection (Rand-OGD) is two-fold:
> - *Data Efficiency:* Models trained on adaptively collected data are able to achieve better performance using the same amount of annotated samples. This is shown in Table 1 and illustrated by the additional results we provided in an *[anonymous Github](https://anonymous.4open.science/r/icml2025-adaptive4robust-CB0F)*. We kindly refer to our response to reviewer 7Pps for further descriptions of these results.
> - *Robustness:* Adaptive data collection is especially suited for distributionally robust learning (minimax objective). In terms of our main metric, the minimum class-wise accuracies for classification and the minimum mAP of each data source for detection, the advantage of the proposed algorithms (and other AL baseline) is significant over Rand-OGD which uses uniform sampling.
>
> **Motivation of Minimax Objective (Q2):** One of the main reasons we consider the minimax objective is that many real-world applications (including our smart intersection example) is *safety-critical*, i.e. we are more concerned with the *worst-case* detection performance among all cameras instead of their average performance. This is essential because the quality of the data and the model accuracy can be largely affected by the specific camera hardware or its geometric location. This motivates our choice of a *minimax objective*, which is standard in robust learning literature. In this context, it is natural to minimize the worst-case loss among all data sources, i.e. $\min_\theta\max_k\mu_k(\theta)$, rather than focusing on the mean loss. In fact, as we show in *Theorem A.4*, algorithms that uniformly collect samples from all data sources (such as Rand-OGD) eventually converge to the optimum of the *mean objective*, i.e. $\min_\theta\frac{1}{K}\sum_{k=1}^K\mu_k(\theta)$, under the same assumptions stated in the paper. We also note that many active learning baselines are also more focused on the mean objective rather than the minimax objective studied by our problem. We believe our proposed framework can fill this important gap of building robust real-world applications under a limited data budget.
>
> **Accounting for Distributional Heterogeneity (Q3):** It is indeed the case that a uniform sampling scheme overlooks the heterogeneous nature of the loss distributions of different data sources, which is why Rand-OGD is only suited for the mean objective. On the other hand, the proposed algorithms like UCB-OGD and Eps-OGD incorporate bandit sampling, which learns the importance of different data distributions from the collected samples in the training set. This provides powerful information about which data source is suffering from worse performance and guides the algorithm to obtain more training samples from it. In this way, algorithms like UCB-OGD and Eps-OGD are able to behave *robustly* and eventually converge to the optimum of the minimax objective (*Theorem 3.1*).
>
> **Our Novelty and Contributions (Q4):** While OGD and UCB are both established algorithms, we believe the combination of these techniques for the studied problem is novel. We want to emphasize that UCB has been proposed for *stationary stochastic bandits*, while our problem is a *non-stationary contextual bandit* where vanilla UCB is not applicable (details in Section 4). In this paper, we prove that, by maintaining an adaptively collected dataset and leveraging its information, UCB (and other stationary bandit algorithms like $\epsilon$-Greedy) can be combined with online optimization algorithms such as OGD to provide *provable minimax regret bounds* despite the non-stationary nature of our problem. We further note that the proposed framework can be easily extended to various bandit and optimization algorithms for different problems at hand. We believe that our paper shows that adaptive data collection with robust theoretical guarantees can be a promising approach applicable to a wide range of tasks.
>
> We appreciate your insights and hope that the answers above could address your concerns. We welcome any further questions or feedback.

---

### Official Review · Reviewer_y1y9 · 2025-03-12

**Overall Recommendation:** 3

**Summary:**

This paper proposes a framework for adaptive data collection and model training considering the multiple data distributions with the fixed annotation budget, where the goal is to come up with an optimized model that can perform well on all distributions. Through the integration of the upper-confidence bound (UCB) for effective sample collection and online gradient descent  (OCG) for the model optimization, the paper proposes a theoretically guaranteed approach called UCB-OGD that shows superior performance on benchmarks compared to existing active learning and random sampling strategies.

**Claims And Evidence:**

1. This paper lacks key baselines in the active learning domain [1,2]. The authors should compare their approach with these baselines in the evaluation. If a direct comparison is not feasible, they should at least discuss these methods in the related works section. Additionally, the authors should explore a sampling strategy based on evidential learning uncertainty and report the performance improvements achieved over this approach [3]. To facilitate comparison, they could consider a pool consisting of samples from all data sources and select samples with the highest uncertainty based on evidential learning uncertainty from that pool.
2. The pure entropy-based sampling strategy is not included in the comparison. The authors should include its performance in Table 1. To select samples, they may consider using a single pool of samples from all the data sources under consideration.
3. The evaluation is limited to computer vision tasks. To increase the impact of this work, the authors should consider natural language tasks (e.g., from the VLM space [4]) and demonstrate how the proposed technique reduces the number of samples from each data source while maintaining comparable performance to using all samples from all data sources.


**References**
1. Ash et al. “Deep Batch Active Learning by Diverse, Uncertain Gradient Lower Bounds”. ICLR2020
2. Kirsch et al. “BatchBALD: Efficient and Diverse Batch Acquisition for Deep Bayesian Active Learning”. NeurIPS2019
3.  Sensoy et al “Evidential Deep Learning to Quantify Classification Uncertainty”. NIPS2018
4. Liu et al. “Visual Instruction Tuning”. NeurIPS2023.

**Essential References Not Discussed:**

NA

**Experimental Designs Or Analyses:**

Yes, I have reviewed the experimental designs and analysis in the main paper (Experimental Results section). The dataset design and analysis appear valid. However, my concerns are as follows: (a) the paper lacks important active learning baselines, (b) the evaluation is limited to computer vision tasks, (c) entropy-based sample selection results are missing from the table, and (d) the active learning (AL) curve is absent. Including an AL curve, with performance (accuracy) on the y-axis and the number of training samples on the x-axis, would provide a clearer comparison of different baselines, including the proposed method, and better illustrate its effectiveness.

**Methods And Evaluation Criteria:**

**Proposed Methodology**: Although the authors provide a detailed theoretical proof for the proposed technique, its novelty is limited. Upper-confidence bound (UCB) and online gradient descent (OGD) are well-established concepts in machine learning, and this paper primarily adapts these techniques for active learning. Specifically, it employs UCB-based uncertainty to guide data source selection.
**Evaluation Criteria**: Strength: This paper has considered multiple computer vision tasks including classification and object detection with multiple datasets.  Weakness: To enhance the impact of this work, authors consider the natural language tasks (e.g., from VLM space [4]) and showcase how this technique helps to reduce the number of samples from each data source with the comparable performance as that of the one considering all sample from all data sources.

**Other Comments Or Suggestions:**

NA

**Other Strengths And Weaknesses:**

I have reiterated Strengths and Weaknesses as follow:

**Strengths**:
1. This paper presents a comprehensive theoretical proof demonstrating the effectiveness of the proposed technique in achieving strong performance across K data sources with effective minimax regret across these sources.
2. The experiments are conducted across multiple tasks, including classification and object detection.
3. Demonstrates the effectiveness of the proposed technique by evaluating its performance on real-world vehicle detection at urban intersections.

**Weaknesses**:
1. This paper lacks key baselines in the active learning domain [1,2]. The authors should compare their approach with these baselines in the evaluation. If a direct comparison is not feasible, they should at least discuss these methods in the related works section. Additionally, the authors should explore a sampling strategy based on evidential learning uncertainty and report the performance improvements achieved over this approach [3]. To facilitate comparison, they could consider a pool consisting of samples from all data sources and select samples with the highest uncertainty based on evidential learning uncertainty from that pool.
2. Although the authors provide a detailed theoretical proof for the proposed technique, its novelty is limited. Upper-confidence bound (UCB) and online gradient descent (OGD) are well-established concepts in machine learning, and this paper primarily adapts these techniques for active learning. Specifically, it employs UCB-based uncertainty to guide data source selection.
3. The pure entropy-based sampling strategy is not included in the comparison. The authors should include its performance in Table 1. To select samples, they may consider using a single pool of samples from all the data sources under consideration.
4. The authors should consider providing an active learning (AL) curve, with the y-axis representing performance (accuracy) and the x-axis showing the number of samples used during training. Comparing different baselines, including their proposed method, would provide a clearer understanding of the technique's effectiveness.
5. The evaluation is limited to computer vision tasks. To increase the impact of this work, the authors should consider natural language tasks (e.g., from the VLM space [4]) and demonstrate how the proposed technique reduces the number of samples from each data source while maintaining comparable performance to using all samples from all data sources.

**References**
1. Ash et al. “Deep Batch Active Learning by Diverse, Uncertain Gradient Lower Bounds”. ICLR2020
2. Kirsch et al. “BatchBALD: Efficient and Diverse Batch Acquisition for Deep Bayesian Active Learning”. NeurIPS2019
 3. Sensoy et al “Evidential Deep Learning to Quantify Classification Uncertainty”. NIPS2018
4. Liu et al. “Visual Instruction Tuning”. NeurIPS2023.

**Questions For Authors:**

Please refer to Other Strengths and Weaknesses section

**Relation To Broader Scientific Literature:**

In the field of active learning, selecting informative samples within a fixed annotation budget is crucial, especially in critical domains like healthcare and road traffic. Unlike prior work, this paper introduces a novel upper-confidence bound (UCB)-based approach for selecting data distributions to enable effective sampling within the given budget. Additionally, the paper provides extensive theoretical analysis to demonstrate how the proposed technique outperforms existing methods. Furthermore, in addition to standard benchmarks, the paper evaluates the approach on a real-world urban intersection dataset, which could serve as an important testbed for future active learning techniques.

**Theoretical Claims:**

I have checked the correctness of the theoretical claims  made in the main paper as well as corresponding proofs in the supplementary materials. I don't see any issue in the theoretical claims.

---

> ### Author Rebuttal · Authors · 2025-04-01
>
> Dear reviewer y1y9,
>
> Thank you for your detailed feedback. We have provided additional experimental results in an *[anonymous Github](https://anonymous.4open.science/r/icml2025-adaptive4robust-CB0F)* and further clarified the results and our contributions below.
>
> **Additional Active Learning (AL) Baselines (weakness 1,3,4):** We add the results of BADGE algorithm (suggested reference [1]) and entropy-based sampling (*Munjal et al., 2022*) on the CIFAR10 dataset, summarized with other algorithms using an AL curve in *[Figure 3](https://anonymous.4open.science/r/icml2025-adaptive4robust-CB0F/fig3-alcurve-cifar10.pdf)* as suggested by the reviewer. In terms of our main metric, the minimum class-wise accuracies are 39.5 for BADGE and 53.0 for entropy-based sampling. The latter is comparable to the results of 52.7 by DBAL and 52.3 by UCB-OGD. We also provide the AL curve on the VOC2012 dataset in *[Figure 4](https://anonymous.4open.science/r/icml2025-adaptive4robust-CB0F/fig4-alcurve-voc2012.pdf)*, which shows superior performance of our algorithm compared to MDN (*Choi et al., 2021*), a SOTA AL baseline. The BatchBALD algorithm (suggested reference [2]) is not tested for now due to its significant computation overhead given the limited time frame. We will also make sure to add the suggested references to the related work of our paper. The main takeaway of these comparisons (also *Table 1* and *Figure 2* of the paper) we want to emphasize is that:
> - AL algorithms are *not necessarily* robust learners. They can be sensitive to initializations, especially for complex tasks like object detection (*Figure 2* of the paper and new *[Figure 4](https://anonymous.4open.science/r/icml2025-adaptive4robust-CB0F/fig4-alcurve-voc2012.pdf)*).
> - On the other hand, our proposed framework can achieve *similar or superior* performance compared to AL baselines for both classification and detection. Meanwhile, the framework is flexible to apply to a wide range of different tasks, since the proposed algorithm solely relies on the losses when making sampling decisions.
>
> **Advanced Sampling Strategies (weakness 1):** While the proposed algorithm can potentially benefit from employing more advanced sampling strategies, provable theoretical guarantees of a general adaptive data collection framework that can incorporate DL models remains elusive even under the simple setup that we discussed in the paper (where the algorithm only selects and optimizes one data source with the highest plausible loss in each round). The outline of the proposed algorithms and the implementations have been set up to resemble the theoretical claims as much as possible, at the price of having more basic algorithmic components (e.g. a simpler sampling scheme). Nonetheless, the experiments already demonstrate the capability of our proposed framework under these basic setups.
>
> **VLM Experiments (weakness 5):** We finetune a SmolVLM-256M-Base [1] model on a subset of VQAv2 [2] dataset and provide the AL curve of per-token accuracy v.s. the number of training samples in *[Figure 5](https://anonymous.4open.science/r/icml2025-adaptive4robust-CB0F/fig5-alcurve-vqav2.pdf)*. We kindly refer to our response to reviewer 7Pps for further details on the experimental setup. Although the time frame is very limited for experiments on a larger scale, we believe the results are able to demonstrate that our proposed framework can be applied to complex tasks like VLM where the training data incorporates various modalities. We will add more detailed experiments in the final version should the paper be accepted.
>
> **Our Novelty and Contributions (weakness 2):** While OGD and UCB are both established algorithms, we believe the combination of these techniques for the studied problem is novel. We want to emphasize that UCB has been proposed for *stationary stochastic bandits*, while our problem is a *non-stationary contextual bandit* where vanilla UCB is not applicable (details in *Section 4*). In this paper, we prove that, by maintaining an adaptively collected dataset and leveraging its information, UCB (and other stationary bandit algorithms like $\epsilon$-Greedy) can be combined with online optimization algorithms such as OGD to provide *provable minimax regret bounds* despite the non-stationary nature of our problem. We further note that the proposed framework can be easily extended to various bandit and optimization algorithms for different problems at hand. We believe that our paper shows that adaptive data collection with robust theoretical guarantees can be a promising approach applicable to a wide range of tasks.
>
> We appreciate your suggestions and insights, and hope that the updates above could address your concerns. We welcome any further questions or feedback.
>
> **References:**
> [1] Marafioti et al., “SmolVLM: Redefining small and efficient multimodal models”. 2025.
> [2] Antol et al., "VQA: Visual Question Answering". ICCV 2015.

---

> > ### Comment · Reviewer_y1y9 · 2025-04-05
> >
> > I would like to thank the authors for providing a comprehensive rebuttal within a short timeframe. I have increased my score to reflect their efforts, particularly in (a) presenting baseline results for other methods (even though some outperform the proposed approach), and (b) providing results for the VLM.

---

### Official Review · Reviewer_7Pps · 2025-03-13

**Overall Recommendation:** 3

**Summary:**

This paper presents a new online framework (UCB-OGD) for data collection and model training in a multi-distributional setting with a constraint on sample labeling. In particular, the purpose UCB-OGD is shown to achieve a sublinear minimax regret with a lower-bound showing algorithmic completion, guaranteeing performance in comparison with traditional active learning methods. Additionally, the paper also presents a well-designed experimental setup to validate the UCB-OGD's effectiveness.

**Claims And Evidence:**

The paper claims theoretical guarantees of their UCB-OGD alongside claims of empirical improvements in a multi-distributional setting with a budget on sample labeling. In this regard, the reviewer found the evidence convincing with both theoretical analysis under traditional bandits and online optimization. Additionally, the experimental evaluations are intuitive and, from the reviewer's perspective, fair.

**Essential References Not Discussed:**

The reviewer is unaware of any necessary references which are omitted.

**Experimental Designs Or Analyses:**

The authors evaluate UCB-OGD using a well-designed proposed setting with three distinct tasks. However, the reviewer would like to see more high resolution datasets in particular for the Classification tasks.

**Methods And Evaluation Criteria:**

The reviewer found the proposed UCB-OGD method to be an intuitive method for addressing the raised active learning problems. Subsequently, the experimental setup is well-designed and the comparative methods chosen are also largely appropriate and fair.

**Other Comments Or Suggestions:**

Minor typos of note:
- line 373 "Detailed of data source assignment is given in Appendix B" -> "Additional details of data source assignment are given in Appendix B"

**Other Strengths And Weaknesses:**

See sections above.

**Questions For Authors:**

See the "experimental designs Or analyses" section.

**Relation To Broader Scientific Literature:**

Algorithmically, UCB-OGD is built on existing algorithms and do not provide too many novel ideas to the broader field. However, the reviewer does find the proposed framework on data collection, bridging ideas from active learning, multi-armed bandits, and online optimization, to be novel and potentially very helpful for the broader active learning field.

**Theoretical Claims:**

The authors provide extensive theoretical analysis that show their proposed UCB-OGD algorithm achieves a sublinear minimax. The paper also provides a lower-bound showing no algorithm can do asymptotically better which matches the known multi-armed bandits theory. The author also clearly state their theoretical assumptions (i.i.d sampling, loss convexity in the model parameters, and bounded loss with Lipschitz continuity). To the reviewer's knowledge the theoretical proofs and claims are compelling with no visible incorrections.

---

> ### Author Rebuttal · Authors · 2025-04-01
>
> Dear reviewer 7Pps,
>
> Thank you for your feedback.
>
> We have provided additional experimental results in an *[anonymous Github](https://anonymous.4open.science/r/icml2025-adaptive4robust-CB0F)*. There are three active learning curves (model performance *v.s.* number of annotated samples) for three different tasks.
>
> **Vision-Language Modeling:** We finetune a SmolVLM-256M-Base model [1] on a subset of VQAv2 dataset [2] for visual question answering using Rand-OGD and UCB-OGD, respectively. We partition the dataset into three data sources based on the type of the questions (i.e. yes/no questions, numerical questions, and descriptive questions). The active learning curve of per-token accuracy *v.s.* the number of training samples is illustrated in *[Figure 5](https://anonymous.4open.science/r/icml2025-adaptive4robust-CB0F/fig5-alcurve-vqav2.pdf)*. Although the time frame is very limited for experiments on a larger scale, we believe the results are able to demonstrate that our proposed framework can be applied to complex tasks like VLM where the training data incorporates various modalities. We will add more detailed experiments in the final version should the paper be accepted.
>
> **Classification and Object Detection:** For the experiments on CIFAR10 and VOC2012 in the paper (*Section 5*), we summarize the performances of the proposed algorithms compared to several active learning baselines in *[Figure 3](https://anonymous.4open.science/r/icml2025-adaptive4robust-CB0F/fig3-alcurve-cifar10.pdf)* and *[Figure 4](https://anonymous.4open.science/r/icml2025-adaptive4robust-CB0F/fig4-alcurve-voc2012.pdf)*, respectively. The main takeaway we want to emphasize is that, while active learning algorithms can be *task-specific* and are *not necessarily* robust learners, our proposed framework is flexible to apply to a wide range of different tasks and is able to behave robustly. The flexibility is due to the fact that the proposed algorithm solely relies on the losses when making sampling decisions.
>
> We appreciate your insights and welcome any further questions or feedback. We will be continuously seeking to add further experiments in support of our theoretical claims.
>
> **References:**
> [1] Marafioti et al., "SmolVLM: Redefining small and efficient multimodal models". 2025.
> [2] Antol et al., "VQA: Visual Question Answering". ICCV 2015.

---

### Decision · Program_Chairs · 2025-05-01

**Decision:**

Accept (poster)

**Comment:**

The reviewers generally found the contribution setting meaningful and the theoretical results well supported. In the rebuttal stage, the reviewers requested adding more baselines, and more experiments in the vision-language setting, and the authors added new experiments to show the versatility of the framework. The reviewers were all in support of acceptance.